# Breaking the Discretization Barrier of Continuous Physics Simulation Learning

**Fan Xu**[1*], **Hao Wu**[2*], **Nan Wang**[3†], **Lilan Peng**[4], **Kun Wang**[5†], **Wei Gong**[1†], **Xibin Zhao**[2†]

[1]University of Science and Technology of China [2]Tsinghua University
[3]Beijing Jiaotong University [4]Southwest Jiaotong University [5]Nanyang Technological University
[*] Equal Contribution, [†] Corresponding author
✉ Main Contact: `markxu@mail.ustc.edu.cn`

## Abstract

The modeling of complicated time-evolving physical dynamics from partial observations is a long-standing challenge. Particularly, observations can be sparsely distributed in a seemingly random or unstructured manner, making it difficult to capture highly nonlinear features in a variety of scientific and engineering problems. However, existing data-driven approaches are often constrained by fixed spatial and temporal discretization. While some researchers attempt to achieve spatio-temporal continuity by designing novel strategies, they either overly rely on traditional numerical methods or fail to truly overcome the limitations imposed by discretization. To address these, we propose `CoPS`, a purely data-driven methods, to effectively model continuous physics simulation from partial observations. Specifically, we employ multiplicative filter network to fuse and encode spatial information with the corresponding observations. Then we customize geometric grids and use message-passing mechanism to map features from original spatial domain to the customized grids. Subsequently, `CoPS` models continuous-time dynamics by designing multi-scale graph ODEs, while introducing a Markov-based neural auto-correction module to assist and constrain the continuous extrapolations. Comprehensive experiments demonstrate that `CoPS` advances the state-of-the-art methods in space-time continuous modeling across various scenarios. The source code is available at `https://github.com/Sunxkissed/CoPS`.

## 1 Introduction

Achieving accurate global modeling and forecasting of a complex time-evolving physical system from a limited number of observations has been a long-standing challenge [5, 52, 11]. This has widespread applications in chaotic physical systems such as geophysics [12, 38], atmospheric science [9, 40], and fluid dynamics [56, 29, 55, 4], *et al*. For instance, in meteorology and oceanography, sensors are often deployed in areas with scarce or non-existent network connectivity. This renders the analysis and modeling of the system a formidable obstacle. From empirical knowledge, dynamical systems can be fully described by complicated and not yet fully elucidated partial differential equations (PDEs) [16]. Traditional numerical methods such as finite element [13] and finite volume [2] methods excessively rely on prior knowledge of essential PDEs and suffer from high time complexity, making it difficult to accommodate the increasing demands for grid granularity. Recently, data-driven approaches partially overcome these by integrating advanced neural networks to directly learn dynamic latent features with high nonlinearity from existing observations or simulations.

Although data-driven methods [18, 31, 45] have the capacity to learn complex relationships from observations, they still have serious limitations in changeable actual scenes. One significant challenge in modeling physical fields from partial observations is that the field may not adhere to a Cartesian

39th Conference on Neural Information Processing Systems (NeurIPS 2025).

grid [34, 1], and convolutional neural networks are inherently not adapted to such structures. When employing graph neural networks [22, 51], the graph topology is often determined by the locations of existing observation points, making the model difficult to generalize to unseen spatial locations. Additionally, some methods use padding or interpolation [6] to enforce spatiotemporal continuity. This may distort the inherent inductive bias of the observed data and significantly increases the computational burden in sparse data scenarios.

Recently, to better explore the space-time continuous formulation from partial observations [19, 53], a few notable works stand out. Example of continuous time and space dynamic system evolution is illustrated in Figure 1. For space-continuous learning, the multiplicative filter network (MFN) [10], combined with implicit neural representations (INRs) [17], has been proposed to incorporate coordinate information through linear

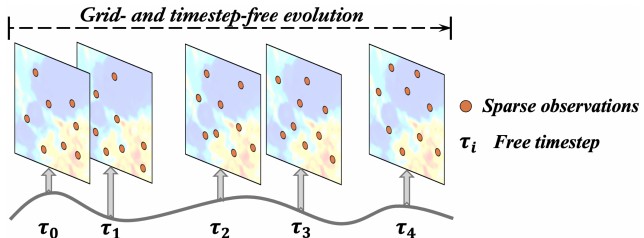

Figure 1: Example of continuous time and space dynamic system evolution. (*Non-discretized*)

Fourier or wavelet functions in an efficient manner. However, this approach requires iterating for hundreds of steps to obtain the corresponding implicit neural representation for new samples, severely lacking real-time applicability. Further, for continuous-time learning, neural ordinary differential equations (Neural ODEs) [7] address the limitation of fixed time extrapolation windows by learning continuous ODEs from discrete data using explicit solvers such as the Euler or Runge-Kutta methods [3, 33]. However, if the system's nonlinearity is high, Neural ODE may struggle to capture the essential features of the complex dynamics. Additionally, Neural ODE is solved through numerical integration, which leads to errors accumulating as time progresses. As a result, conventional Neural ODEs may lead to poor stability in long-term predictions and potentially worse fitting.

To address these, we propose `CoPS`, a purely data-driven methods, to effectively achieve space-time continuous prediction of dynamical systems based on partial observations. Specifically, we employ multiplicative filter networks to fuse and encode spatial information with the corresponding observations. Then we customize geometric grids and use message passing mechanism to map features from original spatial domain to the customized grids. Subsequently, `CoPS` models continuous dynamics by designing multi-scale Graph ODEs, while introducing a local refinement feature extractor to assist and constrain the parametric evolutions. Finally, the predicted latent features are mapped back to original spatial domain through a GNN decoder.

In summary, we make the following key contributions: ① ***Encoding Mechanism.*** We propose a novel encoding approach to integrate partial observations with spatial coordinate information, effectively encoding these features into a customized geometric grid. ② ***Novel Methodology.*** We introduce a multi-scale Graph ODE module to model continuous-time dynamics, complemented by a neural auto-regressive correction module to assist and constrain the parametric evolution, ensuring robust and accurate predictions. ③ ***Superior Performance.*** Our method demonstrates state-of-the-art performance on complex synthetic and real-world datasets, showcasing the possibility for accurate and efficient modeling of intricate dynamical processes and precise long-term predictions.

## 2   Related Work

### 2.1   Deep Learning for Physical Simulations

Recent advancements in deep neural networks have established them as effective tools for addressing the complexities of dynamics modeling [15, 52, 50], demonstrating their ability to efficiently capture the intricacies of high-dimensional systems. Physics-Informed Neural Networks (PINNs) [21, 44, 25], which optimize weights by minimizing the residual loss derived from the PDE, have received considerable attention due to their flexibility in tackling a wide range of data-driven solutions [27, 46]. Recently, neural operators [24, 29, 49, 35] map between infinite–dimensional function spaces by replacing standard convolution with continuous alternatives. Specifically, they utilize kernels in fourier space [29, 28, 42] or graph structure [30, 54, 57, 14] to learn the correspondence from the initial condition to the solution at a fixed horizon. However, most existing methods are limited by static observation grids, restricting their ability to evaluate points outside training grids and

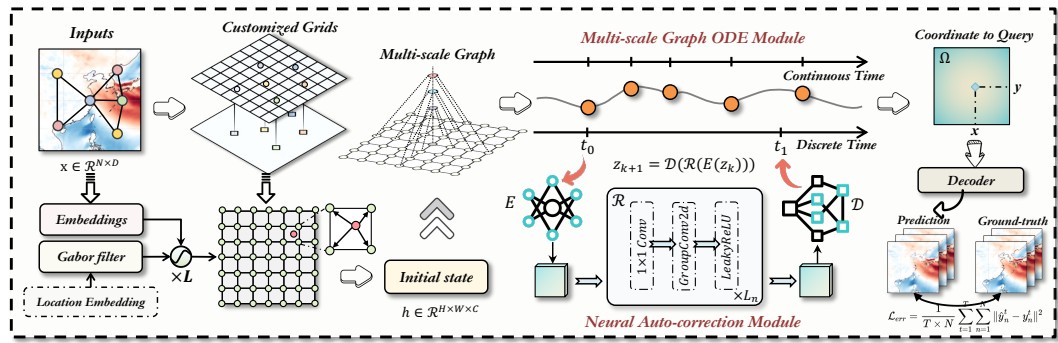

Figure 2: The overview of `CoPS`. *Stage 1:* Employ multiplicative filter network to encode the initial representation, and map it to the customized grids with message passing scheme. *Stage 2:* Model the latent dynamics with multi-scale Graph ODE module and auto-correction module in a continuous-time way. *Stage 3:* Extrapolate results for arbitrary future time step and coordinates.

confining queries to fixed time intervals. In this work, we aim to overcome discretization and achieve spatiotemporal modeling in a continuous way.

## 2.2 Advanced Space-Continuous Modeling

Interpolation has been widely adopted in numerous applications [6, 20, 47], which also holds wide prospects in spatiotemporal modeling. For instance, researchers employed interpolation as a post-processing technique to generate continuous predictions. Recently, MAgNet [5] proposes to predict the dynamical solution after interpolating the observation graph in latent space to new query points. Another approach for space-continuous or grid-independence modeling is a kind of coordinate-based neural networks, called Implicit Neural Representations (INRs) [17, 23]. Typically, INRs take the spatial coordinates as inputs along with other conditions, which can be utilized to enhance real space-continuous modeling and dynamical evolution predicting. DINO [52] overcomes the limitation of failing to generalize to new grids or resolution via a continuous dynamics model of the underlying flow. MMGN [32] learns relevant basis functions from sparse observations to obtain continuous representations after factorizing spatiotemporal variability into spatial and temporal components.

## 3 Methodology

**Problem Definition.** In this work, we focus on modeling spatiotemporal dynamical systems through partial observations. The systems are defined over the temporal domain $\mathcal{T}$ and spatial domain $\Omega$. The observations are recorded at $L$ arbitrary consecutive time points $t_{1:L} := (t_1, \ldots, t_L) \in \mathcal{T}$, and $N$ arbitrary local sensor measurements at locations $x_i \in \Omega, i = (1, \ldots, N)$. To adapt to general real-world scenarios, we assume that all dynamics are learned from the initial observation $u(t_0) = (u(x_1, t_0), \ldots, u(x_N, t_0))^T$. Moreover, the values of time steps and observation locations may vary across different observed trajectories during the inference phase. Thus, for prediction, our goal is to learn a neural function $\mathcal{Q}(\cdot)$, which maps the initial observation at the first time step to future dynamical predictions at arbitrary time steps $t_i$ and locations $x_j$.

$$\mathcal{Q}(u(t_0, x_0); \mathcal{T}, \Omega) \to \mathcal{Y}(t_i, x_j), \quad t_i \in \mathcal{T}, \; x_j \in \Omega. \tag{1}$$

**Framework Overview.** In this section, we introduce the proposed `CoPS`, which achieves the continuous spatiotemporal modeling from partial observations within complicated dynamical systems. We aim to simultaneously overcome the discretization limitations in both the spatial and temporal domains. To better align with the form of numerical solutions, we begin with partial observations of the initial state and apply constraints based on real observations during the continuous evolution process. An overview of our `CoPS` can be seen in Figure 2.

### 3.1 Initial Encoding via Multiplicative Filter Network

**Initial Encoder.** Further, we follow the multiplicative filter network and choose nonlinear Gabor filter [10] $g_\kappa(\cdot)$ to generate a continuous global frequency representation. The formulation is as

follows:

$$g_\kappa\left(x\right) = \exp\left(-\frac{\gamma^{(\kappa)}}{2}\left\|x - \mu^{(\kappa)}\right\|_2^2\right)\sin\left(W_g^{(\kappa)}x + b_g^{(\kappa)}\right),\tag{2}$$

where $\mu^{(\kappa)} \in \mathbb{R}^{d_h}$ and $\gamma^{(\kappa)} \in \mathbb{R}^{d_h \times d_x}$ denote the respective mean and scale term of the $\kappa$-th Gabor filter. Subsequently, the filters are multiplied by the linear transformation of previous layer's embedding and the historical sequence representation $u_i$. The iteration process is defined as follows:

$$\rho_i^1 = g_1\left(x_i\right) \;\Rightarrow\; \rho_i^{(\kappa+1)} = \left(u_i + W_i^\kappa \rho_i^\kappa + b_i^\kappa\right)\odot g_\kappa\left(x\right),\; \text{for}\;\; \kappa = 1,\ldots,\mathcal{M}-1,$$
$$\mathcal{H}_\omega(u_i, x_i) = u_i + W_i^\mathcal{M}\rho_i^\mathcal{M} + b_i^\mathcal{M},\tag{3}$$

where $\odot$ denotes element-wise multiplication, $W_i^\kappa$ and $b_i^\kappa$ are the learnable parameters. Finally, $\mathcal{H}_\omega(u_i, x_i)$ is the obtained vector that contain coordinate information and feature of node $i$.

**Customized Grids and Message Passing Scheme.** For our observational data, the spatial domain is non-discrete. Recent works like MeshGraphNet [34] rely on fixed meshes, which present challenges when generalizing to previously unseen coordinate points. We propose a novel customized grid mapping approach. Specifically, for any arbitrary coordinate, it can be mapped onto a uniform grid structure. Let $p_i = (x_i, y_i)$ denote an arbitrary point in the continuous $2D$ space, we will discover the appropriate grid cell according to it. To enrich the representation, we connect each point $p_i$ to the four vertices $\{v_{i1}, v_{i1}, v_{i3}, v_{i4}\}$ of the associated grid cell.

Based on the encoded initial state $h_i^0 = \mathcal{H}_\omega(u_i, x_i)$ and the customized grids, the next step involves a message-passing scheme that propagates the features from the original continuous space to the customized grids. To further preserve spatial continuity, the scheme encodes and incorporates the relative positions of the connected points during message passing. The transformation is as follows:

$$h_i^{k+1} = h_i^k + \sum_{j\in\mathcal{N}(p_i)} W_{ij}^k\left(h_j^k - h_i^k + \phi(x_i, x_j)\right) + b^k,\tag{4}$$

where $W_{ij}^k$ and $b^k$ are learnable parameters, and $\phi(x_i, x_j)$ denotes the relative position embedding of node $i$ and $j$. In this way, we collaboratively encode both the feature information and rich spatial information onto the customized grids, completing the initial encoding process.

## 3.2 Latent Dynamics Modeling

Further, we view high-dimensional features on the customized grids as latent states and aim to efficiently model complicated dynamics in a continuous-time way.

**Multi-scale Graph ODE.** To capture spatiotemporal dynamics over complex domains, we build a hierarchical graph representation inspired by GraphCast [26] on top of the customized grids. Our objective is to progressively encode both coarse global behaviors and refined local structures. Concretely, let $\{\mathcal{G}^{(s)}\}_{s=1}^S$ be a collection of graphs, where $\mathcal{G}^{(1)}$ coincides with the finest-scale grid obtained from the initial encoding process. Specifically, within each graph $\mathcal{G}^{(s)}$, we connect each node to its spatial neighbors by "jump adjacency" on the underlying $2D$ grid, thus $\mathcal{G}^{(2)},\ldots,\mathcal{G}^{(S)}$ providing graph topologies with longer-range dependencies.

To perform hierarchical message passing, we process each scale independently and then fuse the features using an attention mechanism. For each scale $s$, the message passing is defined as follows:

$$h_i^{'(l,s)} = \text{AGGREGATE}\left(\left\{h_j^{(l-1,s)} : j \in \mathcal{N}^{(s)}(i)\right\}\right), h_i^{(l,s)} = \text{COMBINE}\left(h_i^{(l-1,s)}, h_i^{'(l,s)}\right),\tag{5}$$

where $\mathcal{N}^{(s)}(i)$ denotes the neighbors of node $i$ at $s$-th scale. After propagating messages within each scale, we employ an attention mechanism to fuse the features across scales.

$$h_i^{(l)} = \sum_{s=1}^S \alpha_i^{(s)} h_i^{(l,s)},\quad \alpha_i^{(s)} = (W_{query}h_i^{(l,s)}) \star (W_{key}h_i^{(l-1)}),\tag{6}$$

where $\alpha_i^{(s)}$ denotes the attention weight, $W_{query}$ and $W_{key}$ are learnable parameters, and $\star$ denotes the cosine similarity computation. Building on this hierarchical representation, we parametrize a coupled Graph ODE to model the continuous-time evolution of it's latent state.

$$\frac{dh_i^t}{dt} = \Phi\left(h_1^t, h_2^t, \cdots, h_N^t, \mathrm{G}, \Theta\right).\tag{7}$$

Here, $\Phi$ denotes the multi-scale message passing function, $G$ represents the hierarchical graph structure, and $\Theta$ encapsulates the model parameters. Given the ODE function $\Phi$, the node initial values, and the corresponding contextual vector, the latent dynamics can be solved by any ODE solver like Runge-Kutta [37].

$$h_i^{t_1} \cdots h_i^{t_T} = \text{ODESolve}(\Phi, [h_1^0, h_2^0 \cdots h_N^0]). \tag{8}$$

This formulation allows the model to theoretically obtain the hidden state at any arbitrary timestep, provided that the Neural ODE fits the underlying dynamics well.

**Neural Auto-correction.** Although the multi-scale Graph ODE framework provides a global continuous-time evolution, it still relies on the strong assumption that a learnable ODE exists to model the concrete physical dynamics. Moreover, neural ODEs relying on numerical solvers may fail to capture the system's intrinsic nonlinear features, making it difficult to handle error divergence during evolution. To address this, we introduce a neural auto-correction module that imposes a discrete dynamical regime in the latent space. Thus we can capture complex corrections at selected time steps in a heuristic Markov chain manner.

At each correction step, a convolutional encoder first compresses the current latent representation into a compact state. Subsequently, after the discrete evolution block, a transposed convolution-based decoder reconstructs the refined latent field. This operation acts as a "Markov state observer" [8], learning a discrete single-step mapping in a more compact latent space. The formulation is as follows:

$$E(\cdot) = \sigma(\text{LN}(\text{Conv2d}(\cdot))), \quad D(\cdot) = \text{Tanh}(\text{UnConv2d}(\cdot)). \tag{9}$$

Further, between the encoder and decoder lies a neural block that embodies a discrete transition operator similar to a Markov kernel. Technically, it consists of a $1 \times 1$ *Conv2D*, which strips away extraneous channels to focus on the most critical latent features. Then, it is followed by parallel *GroupConv2D* [41] operators to process these features in multiple subspaces. We utilize $\mathcal{R}$ to represent the combination of the $1 \times 1$ *Conv2D* and the parallel *GroupConv2D*. Then the whole evolution can be formulated as:

$$z(k+1) = \mathcal{D}(\mathcal{R}(E(z(k)))), \tag{10}$$

where $z(k)$ denotes the latent embedding at discrete step $k$. Crucially, this transformation depends only on the current embedding $z(k)$, thus fulfilling a Markov property at each correction step. By encapsulating key spatiotemporal features in $z(k)$ and evolving them into $z(k+1)$ via $\mathcal{R}(\cdot)$, the neural auto-correction module adaptively regularizes the global continuous-time solution. On one hand, it provides a local perspective on how errors can be contained and corrected in each interval. On the other hand, it retains a sufficient memory of the system's evolving states without requiring full historical trajectories.

**Inference Strategy.** During inference, we employ an iterative strategy that combines the multi-scale Graph ODE module and the neural auto-correction module to generate continuous accurate predictions. Starting from the initial state $h_i(t_0)$, the model proceeds as follows:

$\star$ *Continuous Evolution*: The multi-scale Graph ODE module evolves the latent states continuously from $t_k$ to $t_{k+1}$:

$$h_i(t_{k+1}) = h_i(t_k) + \int_{t_k}^{t_{k+1}} f_\theta\left(h_i(t), \{h_j(t)\}_{j \in N(i)}, t\right) dt. \tag{11}$$

$\star$ *Discrete Correction*: At each discrete time step $t_{k+1}$, the neural auto-correction module refines the predicted state:

$$h_i^\omega(t_{k+1}) = h_i(t_{k+1}) + \lambda r_\psi\left(h_i(t_k)\right), \tag{12}$$

where $r_\psi$ denotes the neural auto-correction module, and $\lambda$ is the hyperparameter for balance. The corrected state $h_i^\omega(t_{k+1})$ is used as the initial state for the next time step, and the process repeats until the final prediction time is reached. This inference strategy ensures that the model can generate accurate and stable predictions over long time horizons, while maintaining the continuity and smoothness of the learned dynamics.

### 3.3 Decoder

Upon obtaining the latent states on the customized grids, we project them back to the original continuous spatial domain for final predictions. Specifically, for a query coordinate $q_m = \{x_m, y_m\}$,

we first identify the corresponding grid cell $v_m$ and establish connections to its four vertices, denoted $\{v_{m1}, v_{m2}, v_{m3}, v_{m4}\}$. We then initialize the representation of $q_m$ with a single step Gabor filter transformation $h_m = g_1(q_m)$, and then perform a message-passing update just as in Eq (4). Further, we can obtain the corresponding predictions through a 2-layer MLP decoder $\mathcal{D}(\cdot)$.

### 3.4 Theoretical Analysis

**Theorem 3.1** (***Error Bounding via Hybrid Continuous-Discrete Latent Corrections***). *Consider a spatiotemporal dynamical system whose latent representation $y(t)$ is intended to follow an ideal trajectory $y^*(t)$. The learned dynamics are modeled by a Neural ODE:*

$$\frac{d}{dt}y(t) = \mathcal{F}(y(t), \mathcal{G}, \Theta), \tag{13}$$

*with initial condition $y(t_0)$. The function $\mathcal{F}$, representing the multi-scale graph message passing, is assumed to be $L_\mathcal{F}$-Lipschitz continuous with respect to $y(t)$. Periodic corrections are applied at discrete time steps $t_k = t_0 + k \cdot \Delta t_{\mathrm{corr}}$. Let $y(t_k^-)$ be the state before correction and $y(t_k^+)$ be the state after applying a neural auto-correction operator $\mathcal{C}_\psi$:*

$$y(t_k^+) = y(t_k^-) + \mathcal{C}_\psi(y(t_k^-)). \tag{14}$$

*The error is defined as $e(t) = \|y^*(t) - y(t)\|$. We assume:*

*(a) The combined error from the ODE model discrepancy (vis-à-vis $y^*(t)$) and numerical solver inaccuracies over one interval $\Delta t_{\mathrm{corr}}$ is bounded by $\mathcal{E}_{ODE}$, such that $e(t_{k+1}^-) \leq e^{L_\mathcal{F} \Delta t_{\mathrm{corr}}} e(t_k^+) + \mathcal{E}_{ODE}$.*

*(b) The correction operator $\mathcal{C}_\psi$ reduces the error proportionally and introduces a bounded residual, i.e., $e(t_k^+) \leq \kappa \cdot e(t_k^-) + \delta_\mathcal{C}$, where $0 \leq \kappa < 1$ is a contraction coefficient and $\delta_\mathcal{C} \geq 0$ is a base error from the corrector.*

*Then, for $K = \lfloor(t - t_0)/\Delta t_{\mathrm{corr}}\rfloor$ correction steps, if the effective error amplification per cycle $\alpha_{\mathrm{eff}} = \kappa e^{L_\mathcal{F} \Delta t_{\mathrm{corr}}} < 1$, the error at time $t \approx t_0 + K\Delta t_{\mathrm{corr}}$ is bounded by:*

$$\|e(t)\| \leq \alpha_{\mathrm{eff}}^K \left( e(t_0) - \frac{\kappa \mathcal{E}_{ODE} + \delta_\mathcal{C}}{1 - \alpha_{\mathrm{eff}}} \right) + \frac{\kappa \mathcal{E}_{ODE} + \delta_\mathcal{C}}{1 - \alpha_{\mathrm{eff}}}. \tag{15}$$

*This can be expressed more generally as:*

$$\|e(t)\| \leq C_1 \cdot \alpha_{\mathrm{eff}}^{\lfloor(t-t_0)/\Delta t_{\mathrm{corr}}\rfloor} + C_2, \tag{16}$$

*where $C_1 = \max(0, e(t_0) - C_2)$ or simply $e(t_0)$ for a looser bound, $C_2 = \frac{\kappa \mathcal{E}_{ODE} + \delta_\mathcal{C}}{1 - \alpha_{\mathrm{eff}}}$ is the asymptotic error floor, and $\alpha_{\mathrm{eff}} \in [0, 1)$ quantifies the error decay rate per correction cycle, dependent on the system's Lipschitz constant, the correction interval, and the corrector's efficacy. This bound demonstrates that the error converges to a non-zero floor $C_2$ if $\alpha_{\mathrm{eff}} < 1$.*

## 4 Experiment

### 4.1 Experimental Settings

**Datasets.** To comprehensively illustrate the property and efficacy of the extrapolations obtained from CoPS, we conduct experiments on diverse synthetic and real-world datasets. ❖ **For synthetic datasets**, we first choose *Navier-Stokes* [29] and *Rayleigh–Bénard Convection* [43], which are directly generated by numeric PDE solvers. Then, we select *Prometheus* [48], which is a large-scale combustion dataset simulated with industrial software. ❖ **For real-world datasets**, we choose *WeatherBench* [36], a dataset for weather forecasting and climate modeling. We also select *Kuroshio* [49], which provides vector data of sea surface stream velocity from the Copernicus Marine Service. See Appendix A for detailed descriptions of all these datasets.

**Baselines.** We evaluate our model against three baselines representing the state-of-the-art in continuous modeling. **MAgNet** [5] employs an "Encode-Interpolate-Forecast" scheme. Specifically, MAgNet employs the nearest neighbor interpolation technique to generalize to new query points.

Subsequently, it forecasts the evolutionary trends by leveraging a GNN-based message passing neural network. **DINO** [52] learns PDE's flow to forecast its dynamical evolution by leveraging a spatial implicit neural representation modulated by a context vector and modeling continuous-time evolution with a learned latent ODE. This is the closest approach to our method. **ContiPDE** [39] formulates the task as a double observation problem. It utilizes recurrent GNNs to roll out predictions of anchor states from the IC, and employs spatiotemporal attention observer to estimate the state at the query position from these anchor states. We utilize the official implementation for all models and tune their experimental settings to follow the requirements of our tasks.

**Tasks.** We assess the performance of `CoPS` across diverse forecasting tasks to evaluate their efficacy in various scenarios. We use the mean squared error (MSE) as the performance measurement. *(i)* **Sparse Flexibility** - Gauging the effectiveness of predicting global field evolution based on observations with diverse degrees of sparsity (subsampling ratio of 25%, 50%, 75%). *(ii)* **Time Flexibility** - Evaluating our model's performance in extrapolating beyond the training horizon. Here we design two experimental setups: long-term extrapolation beyond the training horizon, and continuous-time prediction for intermediate points between discrete time steps. *(iii)* **Resolution Generalization** - Investigating the performance of generalizing to new resolution (up or down). *(iv)* **Super-resolution** - Investigating the model's effectiveness of super-resolution query and extrapolation. *(v)* **Noise Robustness** - Exploring the robustness of our model against varying noise ratios. *(vi)* **Ablation Study** - Investigating contributes of each key component to the performance of `CoPS`.

**Implementation.** For synthetic data like Navier-Stokes, the train and test sets differ only by their initial conditions. The samples are partitioned in a 7:2:1 ratio into training, validation, and test sets. For real-world data like WeatherBench, we use the historical ERA5 global atmospheric reanalysis data. The data was partitioned strictly by date to prevent any data leakage from the future into the training process. Specifically, the train set uses data from 1979-2018, the validation set from 2019, and the test set from 2020-2022. To ensure fairness, we use partial observations of the initial state as input and supervise the training with future real observations at discrete time steps (only using observed future values). In evaluating spatial continuity, we perform subsampling of the full initial state at rates of {25%, 50%, and 75%}, assessing both observed and unobserved points. For temporal continuity, we evaluate three criteria based on different subsampling rates: predictions within the training horizon, long-term predictions beyond the training horizon, and continuous-time predictions for intermediate points between discrete time steps. More details are illustrated in Appendix C.

## 4.2 Main Results

**Space Flexibility.** To evaluate the spatial querying ability of our method, we adjust the number of available measurement points by using different proportions of observation points (25%, 50%, 75%) in the training data. We report the MSE compared to the ground truth on four datasets, as shown in Table 1. From the table, we see that our method (Ours) significantly outperforms the baseline methods on all datasets and at all observation sparsity levels. On the Navier-Stokes dataset, with only 25% of the observation data, our method achieves MSE of 2.964E-03 and 5.743E-03 under the In-s and Ext-s settings, which are much better than ContiPDE (5.456E-03 and 9.523E-03) and DINo (1.074E-02 and 2.537E-02). As the observation ratio increases to 50% and 75%, our model's performance further improves. The MSE for In-s decrease to 2.253E-03 and 1.464E-03, and for Ext-s decrease to 4.571E-03 and 2.872E-03. On the Kuroshio dataset, even with only 25% observation data, our method achieves MSE of 1.285E-03 and 2.381E-03 under In-s and Ext-s, significantly better than other methods. For example, ContiPDE's MSE are 2.352E-03 and 3.763E-03, and DINo's are 3.737E-03 and 5.923E-03. As the observation ratio increases, our method's MSE further decreases, and its accuracy becomes higher. On the Prometheus and WeatherBench datasets, our method also performs excellently. Especially on the WeatherBench dataset, when the observation ratio is 75%, our method achieves MSE of 3.585E-03 and 8.472E-03 under In-s and Ext-s, much lower than other baseline methods. These results show that our method effectively recovers the global scene from partial observations on different datasets and observation sparsity levels. It demonstrates robustness in handling sparse data and spatial generalization.

**Time Flexibility.** To evaluate our method's performance in temporal extrapolation, we train and evaluate the model with observation subsampling ratio of 50% and 100%. We use three extrapolation modes: (1) Extrapolation within the discrete training range (In-t); (2) Extrapolation beyond the discrete training range (Ext-t); (3) Extrapolation at intermediate points between time steps (Con-t).

Table 1: To evaluate the spatial inquiry power of our method, we vary the number of available measurement points in the data for training from 25%, 50% and 75% amount of observations. We report MSE compared to the ground truth solution.

| Model | | NAVIER-STOKES | | | KUROSHIO | | | PROMETHEUS | | | WEATHERBENCH | | |
|---|---|---|---|---|---|---|---|---|---|---|---|---|---|
| | | s=25% | s=50% | s=75% | s=25% | s=50% | s=75% | s=25% | s=50% | s=75% | s=25% | s=50% | s=75% |
| MAGNET | IN-S | 3.164E-02 | 2.775E-02 | 2.268E-02 | 7.104E-03 | 6.523E-03 | 4.845E-03 | 1.694E-02 | 1.273E-02 | 1.186E-02 | 3.635E-02 | 3.028E-02 | 2.483E-02 |
| | EXT-S | 5.846E-02 | 4.285E-02 | 3.114E-02 | 9.464E-03 | 8.173E-03 | 7.518E-03 | 2.775E-02 | 2.322E-02 | 1.724E-02 | 5.356E-02 | 4.972E-02 | 3.295E-02 |
| DINO | IN-S | 1.074E-02 | 9.564E-03 | 7.554E-03 | 3.737E-03 | 3.332E-03 | 2.884E-03 | 1.223E-02 | 8.005E-03 | 5.657E-03 | 2.365E-02 | 2.186E-02 | 1.922E-02 |
| | EXT-S | 2.537E-02 | 1.764E-02 | 9.248E-03 | 5.923E-03 | 5.271E-03 | 4.487E-03 | 1.784E-02 | 1.246E-02 | 8.324E-03 | 3.823E-02 | 2.746E-02 | 2.588E-02 |
| CONTIPDE | IN-S | 5.456E-03 | 4.176E-03 | 3.825E-03 | 2.352E-03 | 1.947E-03 | 1.503E-03 | 8.462E-03 | 6.084E-03 | 4.763E-03 | 1.542E-02 | 1.294E-02 | 1.056E-02 |
| | EXT-S | 9.523E-03 | 7.975E-03 | 6.231E-03 | 3.763E-03 | 2.531E-03 | 2.084E-03 | 1.125E-02 | 8.355E-03 | 6.674E-03 | 2.172E-02 | 1.653E-02 | 1.474E-02 |
| OURS | IN-S | **2.964E-03** | **2.253E-03** | **1.464E-03** | **1.285E-03** | **7.253E-04** | **5.253E-04** | **5.176E-03** | **3.722E-03** | **2.746E-03** | **8.766E-03** | **5.249E-03** | **3.585E-03** |
| | EXT-S | **5.743E-03** | **4.571E-03** | **2.872E-03** | **2.381E-03** | **1.577E-03** | **9.272E-04** | **8.264E-03** | **5.142E-03** | **4.287E-03** | **1.769E-02** | **1.056E-02** | **8.472E-03** |

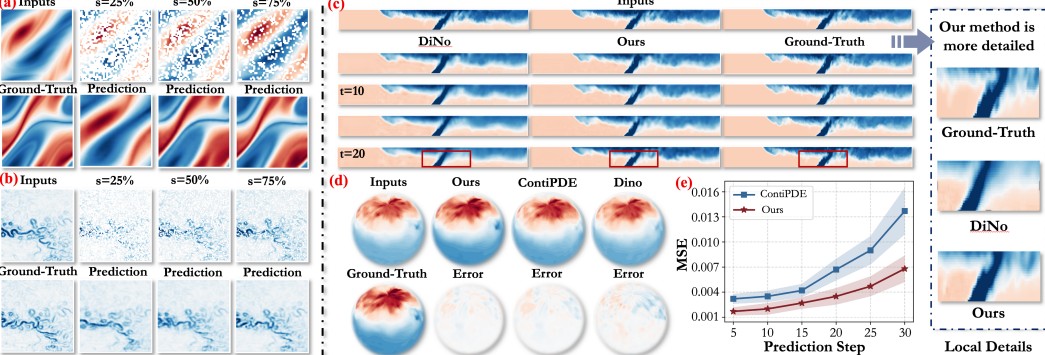

Figure 3: Figure(a) and (b) shows the perdiction performance based on observations with **diverse ratio of subsampling** (25%, 50%, 75%) on the Navier-Stokes and Kuroshio dataset. Figure(c) demonstrates **long-term extrapolation beyond the training horizon** on the Prometheus dataset, and Figure(e) shows the evolution of prediction errors over time steps, revealing the increasing error with longer prediction steps. Figure(d) illustrates **continuous-time prediction** for intermediate points between discrete time steps on the WeatherBench dataset.

We report the mean squared error (MSE) on the Navier-Stokes, Rayleigh–Bénard Convection, and Prometheus datasets, as shown in Table 2. From Table 2, we see that our method achieves the best performance on mostly all datasets and extrapolation settings. For example, on the Navier-Stokes dataset with a subsampling ratio of 50%, our method achieves MSE of 2.832E-03 (In-t), 5.764E-03 (Ext-t), and 4.135E-03 (Con-t). These are significantly better than ContiPDE (5.054E-03, 8.342E-03, 6.447E-03) and DINo (7.651E-03, 1.074E-02, 9.971E-03). When the subsampling ratio increases to 100%, the performance improves further. Our method's MSE decrease to 1.132E-03 (In-t), 2.364E-03 (Ext-t), and 1.735E-03 (Con-t). On the Rayleigh–Bénard Convection dataset, our method also achieves MSE of 6.522E-04 (In-t) and 1.327E-03 (Ext-t), significantly better than other baseline methods. This shows that our method can make accurate predictions within the discrete training range. It can also effectively extrapolate to time points beyond the training range. When we perform continuous-time prediction at intermediate points between time steps (Con-t), our method also performs excellently. It achieves the smallest prediction error. Overall, these results demonstrate the strong capability of our method in continuous-time modeling. It provides accurate and reliable predictions under different temporal extrapolation settings.

**Visualization and Analysis.** Figure 3 provides compelling visual evidence for the superior performance of our proposed `CoPS` across a range of challenging tasks. Specifically, Figure 3 (a) and (b) demonstrate that `CoPS` can effectively model the entire physical field and learn the evolution trends of unobserved points only with partial observations. When the subsampling ratio reaches 75%, our model can achieve a precise prediction on Navier-Stokes and Kuroshio datasets. As observed in Figure 3 (c) and (e), our method significantly outperforms DINO and ContiPDE in long-term forecasting on the Prometheus dataset by effectively suppressing error accumulation, thereby maintaining high-fidelity details where baselines exhibit blurring. This robust temporal modeling is further evidenced in Figure 3 (d), where `CoPS` accurately predicts intermediate states between discrete time steps. This showcases its ability to perform true, dynamically consistent integration rather than simple interpolation, confirming its superior continuous-time modeling capabilities.

Table 2: We evaluate the temporal extrapolation performance of our method. Models are trained and evaluated with observation subsampling ratio of 50% and 100%. We employ three kinds of extrapolation, containing: (1) extrapolation within discrete training horizon (In-t); (2) extrapolation exceeding discrete training horizon (Ext-t); and (3) extrapolation of intermediate points between time steps (Con-t). We report MSE compared to the ground truth solution.

| | NAVIER-STOKES | | | RAYLEIGH–BÉNARD CONVECTION | | | PROMETHEUS | | |
|---|---|---|---|---|---|---|---|---|---|
| | IN-T | EXT-T | CON-T | IN-T | EXT-T | CON-T | IN-T | EXT-T | CON-T |
| | *s* = 50% *subsampling ratio* | | | | | | | | |
| MAGNET | 1.253E-02 | 2.601E-02 | – | 9.421E-03 | 1.335E-02 | – | 1.138E-02 | 1.824E-02 | – |
| DINO | 7.651E-03 | 1.074E-02 | 9.971E-03 | 5.748E-03 | 8.472E-03 | 7.342E-03 | 8.321E-03 | 1.117E-02 | 8.852E-03 |
| CONTIPDE | 5.054E-03 | 8.342E-03 | 6.447E-03 | 3.744E-03 | 3.814E-03 | 2.908E-03 | 5.435E-03 | 8.154E-03 | 6.637E-03 |
| OURS | **2.832E-03** | **5.764E-03** | **4.135E-03** | **1.526E-03** | **2.328E-03** | **1.765E-03** | **3.374E-03** | **6.678E-03** | **5.355E-03** |
| | *s* = 100% *subsampling ratio* | | | | | | | | |
| MAGNET | 7.429E-03 | 1.273E-02 | – | 4.837E-03 | 7.235E-03 | – | 7.938E-03 | 1.024E-02 | – |
| DINO | 4.352E-03 | 7.374E-03 | 5.271E-03 | 2.248E-03 | 4.872E-03 | 3.742E-03 | 5.721E-03 | 8.217E-03 | 6.288E-03 |
| CONTIPDE | 2.655E-03 | 4.742E-03 | 3.847E-03 | 1.244E-03 | 3.214E-03 | 2.308E-03 | 3.835E-03 | 6.554E-03 | 4.037E-03 |
| OURS | **1.132E-03** | **2.364E-03** | **1.735E-03** | **6.522E-04** | **1.327E-03** | **1.061E-03** | **2.877E-03** | **5.174E-03** | **3.852E-03** |

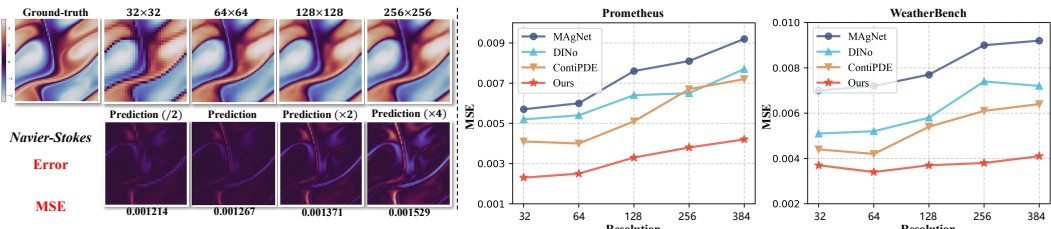

Figure 4: *Left* shows the inference performance for super-resolution on Navier-Stokes dataset. *Right* demonstrates the resolution generalization capability compared with MAgNet, DINo, and ContiPDE.

## 4.3 Model Analysis

**Analysis of Super-resolution.** In the super-resolution experiment presented in Figure 4 (*left*), the results illustrate the inference performance on the Navier-Stokes dataset across resolutions of $32 \times 32$, $64 \times 64$, $128 \times 128$, and $256 \times 256$. The results consistently demonstrate improved prediction quality as resolution increases. Compared to baseline models (MAgNet, DINo, and ContiPDE), the proposed model effectively reduces prediction error at high resolutions, maintaining a lower MSE and providing clearer, more accurate predictions of the original Navier-Stokes patterns.

**Resolution Generalization.** As observed in Figure 4 (*right*), the folding diagrams depict the resolution generalization capabilities for both Prometheus and WeatherBench datasets. The proposed `CoPS` consistently outperforms the comparative models (MAgNet, DINo, ContiPDE) at all resolution levels, achieving the lowest MSE value. This result highlights `CoPS`'s robustness and effectiveness in handling resolution variations. The findings emphasize the model's ability to generalize across different resolutions, which is crucial for applications requiring detailed and precise predictions in complex dynamic systems.

**Ablation Study.** To evaluate the contribution and importance of each component in the proposed `CoPS`, we conduct corresponding ablation experiments, and use MSE error as metric. Our model variants are as follows: ❶ `CoPS w/o MFN`, we remove the multiplicative filter network and use an MLP. ❷ `CoPS w/o MGO`, we remove the multi-scale Graph ODE module. ❸ `CoPS w/o

Table 3: Ablation study on Kuroshio. (Report MSE)

| | IN-S / IN-T | EXT-S / IN-T | IN-S / EXT-T | EXT-S / EXT-T |
|---|---|---|---|---|
| W/O MFN | 1.027E-03 | 1.946E-03 | 2.458E-03 | 3.149E-03 |
| W/O MGO | 1.594E-03 | 2.357E-03 | 3.053E-03 | 4.175E-03 |
| W/O NAC | 1.428E-03 | 2.216E-03 | 2.742E-03 | 3.657E-03 |
| OURS | **7.253E-04** | **1.577E-03** | **2.161E-03** | **2.938E-03** |

NAC**, we remove the neural auto-correction module. Table 3 shows the comprehensive results of our ablation experiment. The results of the ablation experiments show that removing any component results in a decrease in predictive performance, further proving the critical role of these components in `CoPS` framework. Specifically, the performance declines after removing the multi-scale Graph ODE module is particularly evident, proving its importance in capturing internal dynamic representations.

# 5 Conclusion

In this paper, we introduce `CoPS`, a novel data-driven framework for modeling continuous spatiotemporal dynamics from partial observations. To overcome discretization, we first customize geometric grids and employ multiplicative filter network to fuse and encode spatial information with the corresponding observations. After encoding to the concrete grids, `CoPS` model continuous-time dynamics by designing multi-scale graph ODEs, while introducing a Marcov-based neural auto-correction module to assist and constrain the continuous extrapolations. Extensive experiments on both synthetic and real-world datasets demonstrate the superior performance of `CoPS`, which underlines the potential to provide a robust method for accurate long-term predictions in face of sparse and unstructured data.

## Acknowledgement

This work was partially supported by the NSFC Program (No. 62202042, No. 6212780016); in part by the Supported by the Fundamental Research Funds for the Central Universities (No. 2024JBMC031); in part by the Aeronautical Science Foundation of China (No. ASFC-2024Z0710M5002; in part by the by the OpenFund of Advanced Cryptography and System Security Key Laboratory of Sichuan Province(No. SKLACSS-202312); the Guangdong S&T Programme (No. 2024B0101030002), the Ministry of Industry and Information Technology of China, the National Key Research and Development Program of China (No. 2023YFB3307500), and the Science and Technology Innovation Project of Hunan Province (No. 2023RC4014).

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

# A  Detailed Description of Datasets

**Navier-Stokes Equations.**   Navier-Stokes Equations [29] describe the motion of fluid substances such as liquids and gases. These equations are a set of partial differential equations that predict weather, ocean currents, water flow in a pipe, and air flow around a wing, among other phenomena. The equations arise from applying Newton's second law to fluid motion, together with the assumption that the fluid stress is the sum of a diffusing viscous term proportional to the gradient of velocity, and a pressure term. The equations are expressed as follows:

$$\rho \left( \frac{\partial u}{\partial t} + u \cdot \nabla u \right) = -\nabla p + \nabla \cdot \tau + f,$$
$$\frac{\partial \rho}{\partial t} + \nabla \cdot (\rho u) = 0, \tag{17}$$
$$\frac{\partial(\rho E)}{\partial t} + \nabla \cdot ((\rho E + p)u) = \nabla \cdot (\tau \cdot u) + \nabla \cdot (k\nabla T) + \rho f \cdot u,$$

where $u$ denotes the velocity field, $\rho$ represents the density of the fluid, $p$ is the pressure, $\tau$ is the viscous stress tensor, given by $\mu(\nabla u + (\nabla u)^T) - \frac{2}{3}\mu(\nabla \cdot u)\mathbf{I}$. $E$ is the total energy per unit mass, $E = e + \frac{1}{2}|u|^2$, $e$ is the internal energy per unit mass, $T$ denotes the temperature, and $k$ represents the thermal conductivity. All simulations were generated from the Navier-Stokes equation with a constant Reynolds number of 1e-5. We utilized a total of 1200 independent simulation samples.

**Rayleigh-Bénard Convection.**   Rayleigh-Bénard Convection [43] is generated using the Lattice Boltzmann Method to solve the 2-dimensional fluid thermodynamics equations for two-dimensional turbulent flow. The general form of the equations is expressed as:

$$\nabla \cdot u = 0,$$
$$\frac{\partial u}{\partial t} + (u \cdot \nabla)u = -\frac{1}{\rho_0}\nabla p + \nu\nabla^2 u + [1 - \alpha(T - T_0)]X, \tag{18}$$
$$\frac{\partial T}{\partial t} + (\mathbf{u} \cdot \nabla)T = \kappa\nabla^2 T,$$

where $g$ is the gravitational acceleration, $\mathbf{X}$ is the acceleration due to the body-force of the fluid parcel, $\rho_0$ is the relative density, $T$ represents temperature, $T_0$ is the average temperature, $\alpha$ denotes the coefficient of thermal expansion, and $\kappa$ is the thermal conductivity coefficient. The simulation parameters for the dataset are as follows: Prandtl number = 0.71, Rayleigh number = $2.5 \times 10^8$, and the maximum Mach number = 0.1.

**Prometheus.**   Prometheus [48] is a large-scale, out-of-distribution (OOD) fluid dynamics dataset designed for the development and benchmarking of machine learning models, particularly those that predict fluid dynamics under varying environmental conditions. This dataset includes simulations of tunnel and pool fires (represented as Prometheus-T and Prometheus-P in experiments), encompassing a wide range of fire dynamics scenarios modeled using fire dynamics simulators that solve the Navier-Stokes equations. Key features of the dataset include 25 different environmental settings with variations in parameters such as Heat Release Rate (HRR) and ventilation speeds. In total, the Prometheus dataset encompasses 4.8 TB of raw data, which is compressed to 340 GB. It not only enhances the research on fluid dynamics modeling but also aids in the development of models capable of handling complex, real-world scenarios in safety-critical applications like fire safety management and emergency response planning.

**WeatherBench.**   WeatherBench [36] is a benchmark dataset designed for the evaluation and comparison of machine learning models in the context of medium-range weather forecasting. It is intended to facilitate the development of data-driven models that can improve weather prediction, particularly in the range from 1 to 14 days ahead. The dataset consists of historical weather data from multiple atmospheric variables, including temperature, pressure, humidity, wind speed, and geopotential height, at various global locations. The data is derived from the ERA5 reanalysis, which provides hourly estimates of the atmosphere's state at a resolution of 31 km for the period from 1950 to present. Its primary goal is to serve as a benchmarking tool to compare the performance of machine learning-based models against traditional numerical weather prediction methods. By focusing on

data-driven techniques, it aims to push forward the development of models that can predict weather patterns in an interpretable and scalable manner, and thus contribute to improving operational weather forecasting systems.

**Kuroshio.** Kuroshio [49] is a dataset designed to study the dynamics of the Kuroshio Current, a major oceanic current in the Pacific Ocean that flows from the east coast of Taiwan and Japan, obtained from the Copernicus Marine Environment Monitoring Service (CMEMS). Key features of the dataset include high temporal and spatial resolution measurements, covering daily and monthly intervals, which allow for in-depth studies on the seasonal and inter-annual variability of the Kuroshio Current. It also includes data on associated oceanic phenomena such as eddy formation, upwelling, and interactions with surrounding currents like the Tsushima Current. This dataset covers the period from 1993 to 2024, and we use data from 1993–2020 for training, while data from 2021–2024 for validation and testing.

# B  Pseudocode of CoPS

---
**Algorithm 1** Algorithm workflow of CoPS
---

1: **Input:** Initial observations $U_0 = \{(u(x_i, t_0), x_i)\}_{i=1}^{N_{obs}}$; Query set $Q_{set}$; Model parameters $\Theta_{all}$; Hyperparameters $\Delta t_{corr}, \lambda_c$.

2: **Output:** Predictions $\hat{Y} = \{\hat{u}(t_q, x_q)\}$ for $(t_q, x_q) \in Q_{set}$.

3: /* **Stage 1: Initial Encoding and Grid Mapping** */

4: $\quad \{h_i(t_0)\} \leftarrow \text{Encoder}_{\Theta_{enc}}(U_0)$     ▷ Encode observations to point embeddings     ▷ Eq. 2, 3

5: $\quad z(t_0)^+ \leftarrow \text{GridMapper}_{\Theta_{map}}(\{h_i(t_0)\})$     ▷ Map to initial latent grid state     ▷ Eq. 4

6: /* **Stage 2: Iterative Latent Dynamics Prediction with Correction** */

7: $\quad z_k^+ \leftarrow z(t_0)^+; t_k \leftarrow t_0; \mathcal{Z}_{traj} \leftarrow \{(t_0, z(t_0)^+)\}$

8: **while** $t_k < \max_{(t_q, \cdot) \in Q_{set}}\{t_q\}$ **do**

9: $\quad\quad t_{next\_corr} \leftarrow t_k + \Delta t_{corr}$

10: $\quad\quad z_{k+1}^- \leftarrow \text{ODESolve}(\Phi_{\Theta_{ode}}, z_k^+, [t_k, t_{next\_corr}])$     ▷ Continuous evolution     ▷ Eq. 7, 8, 11

11: $\quad\quad$ Store evolution from $z_k^+$ to $z_{k+1}^-$ in $\mathcal{Z}_{traj}$.

12: $\quad\quad z_{k+1}^+ \leftarrow z_{k+1}^- + \lambda_c \cdot A_{\Theta_{ac}}(z_k^+)$     ▷ Apply discrete correction     ▷ Eq. 9, 10, 12

13: $\quad\quad z_k^+ \leftarrow z_{k+1}^+; t_k \leftarrow t_{next\_corr}$

14: **end while**

15: /* **Stage 3: Decoding to Query Locations** */

16: $\hat{Y} \leftarrow \{\}$

17: **for each** query $(t_q, x_q)$ in $Q_{set}$ **do**

18: $\quad$ Identify grid cell $v_{cell}$ and its vertices $\{v_j\}$ for $x_q$.

19: $\quad h_{q,init} \leftarrow \text{GaborFilter}(x_q)$     ▷ Initialize query point representation

20: $\quad h_{q,grid} \leftarrow \text{MessagePass}(\{z(t_q)_{v_j}\}, h_{q,init}, x_q, \{x_{v_j}\}; \Theta_{map})$ ▷ Refine using grid states ▷ Eq. 4 variant

21: $\quad \hat{u}(t_q, x_q) \leftarrow \text{MLPDecoder}(h_{q,grid}; \Theta_{dec})$

22: $\quad$ Add $\hat{u}(t_q, x_q)$ to $\hat{Y}$.

23: **end for**

24: /* **Training:** Parameters $\Theta_{all} = \{\Theta_{enc}, \Theta_{map}, \Theta_{ode}, \Theta_{ac}, \Theta_{dec}\}$ are learned end-to-end by minimizing prediction error. */

25: **return** $\hat{Y}$

---

# C  Details of Implementation

To ensure fairness, we conducted all experiments on an NVIDIA-A100 GPU using the MSE loss over 200 epochs. We used Adam optimizer with a learning rate of $10^{-3}$ for training. The batch size was set to 16.

## C.1 Model hyper-parameters

The core architecture of the model consists of three main modules: the spatial information encoder, the multi-scale graph ODE module, and the neural auto-regressive correction module. The spatial information encoder employs Multiplicative Filter Networks to fuse partial observations with spatial coordinate information and encode them into customized geometric grids. For regular arranged datasets, the customized grid is set to the general resolution. For irregular arranged datasets, it is uniformly set to 128×128. The MFN is configured with 5 layers and uses ReLU as the activation function to ensure efficient feature extraction. The hidden dimension is set to 128. Then the multi-scale graph ODE module utilizes a Runge-Kutta ode solver for numerical integration, with a time step size of 0.25 to ensure accurate modeling of temporal continuity. Further, the neural auto-regressive correction module performs corrections per integer time step. For this module, the Conv2d layer is downsampled to half the resolution, while the UpConv2d layer restores the grid to the original resolution. The parallel GroupConv2d operations are implemented with filter sizes of 3×3, 5×5, and 7×7. For inference, the correction weight $\lambda$ is set to 0.5 to balance correction strength and model stability. Finally, in the decoder, we use a single step Gabor filter transformation to initial the features of query coordinates, and perform a 2-layers message-passing update to obtain the corresponding predictions.

## C.2 Baseline implementation

**MAgNet** [5]. We utilize the official implementation of MAgNet, utilizing a graph neural network variant of the model. The configuration involves five message-passing steps. The architecture of all MLPs includes four layers, with each layer containing 128 neurons. Additionally, we set the dimensionality of the latent state at 128.

**DINo** [52]. We utilize the official implementation of DINo. Specifically, the encoder features an MLP, comprising three hidden layers with 512 neurons each, and Swish non-linearities. The dimension of each hidden layer is set to 100. Similarly, the dynamics function is realized through an MLP, which also includes three hidden layers, each containing 512 neurons and employing Swish activation function. The decoder is constructed with three layers, each with a capacity of 64 channels.

**ContiPDE** [39]. ContiPDE formulates the task as a double observation problem. It utilizes recurrent GNNs to roll out predictions of anchor states from the IC, and employs spatio-temporal attention observer to estimate the state at the query position from these anchor states. First, it utilize a two-layered MLP with 128 neurons, with Swish activation functions to encode features form sparse observations. Further, it uses a two-layered gated recurrent unit with a hidden vector of size 128, and a two-layered MLP with 128 neurons activated by the Swish function to realize the recurrent GNNs. Finally, it employs multi-head attention mechanism to decode and utilizes multi-head attention mechanism to realize continuous query.

# D  Broader impacts

The `CoPS` framework has broad positive impacts in several areas. First, it provides crucial technical support in scientific research, especially in geophysics, atmospheric science, and fluid dynamics. By performing continuous spatiotemporal simulations from sparse observational data, this method achieves high-precision predictions with limited sensor numbers, significantly improving the accuracy and timeliness of weather forecasts and disaster warnings.

# E  Limitations of This Study

While our proposed `CoPS` framework demonstrates significant advancements in modeling continuous spatiotemporal dynamics from partial observations, we acknowledge several limitations that provide avenues for future research. The first is the computational cost of multi-scale graph ODEs. It can be computationally intensive, especially for very fine-grained customized grids or a large number of scales. Future work could explore more efficient graph neural network architectures or adaptive scaling mechanisms to mitigate this. The second is the assumption of Markov property in auto-correction. While this simplifies the model and makes it computationally tractable, real-world

physical systems might exhibit longer-range temporal dependencies that are not fully captured by this discrete correction mechanism.

# F   Proofs of Theorem 3.1

Let $e_k^- = e(t_k^-)$ denote the error just before the $k$-th correction (or at the start of the $k$-th ODE integration interval), and $e_k^+ = e(t_k^+)$ denote the error just after the $k$-th correction. The initial error is $e_0^- = e(t_0)$.

From assumption (b), after a correction at time $t_k$:
$$e_k^+ \le \kappa e_k^- + \delta_{\mathcal{C}}. \tag{19}$$
From assumption (a), after ODE evolution from $t_k$ to $t_{k+1}$:
$$e_{k+1}^- \le e^{L_{\mathcal{F}}\Delta t_{\text{corr}}} e_k^+ + \mathcal{E}_{\text{ODE}}. \tag{20}$$
We want to establish a recurrence relation for $e_k^+$. Substitute $Eq(2)$ (with $k$ replaced by $k+1$ for $e_{k+1}^-$) into $Eq(1)$ (for $e_{k+1}^+$):
$$e_{k+1}^+ \le \kappa e_{k+1}^- + \delta_{\mathcal{C}} \le \kappa(e^{L_{\mathcal{F}}\Delta t_{\text{corr}}} e_k^+ + \mathcal{E}_{\text{ODE}}) + \delta_{\mathcal{C}} = \kappa e^{L_{\mathcal{F}}\Delta t_{\text{corr}}} e_k^+ + \kappa\mathcal{E}_{\text{ODE}} + \delta_{\mathcal{C}}. \tag{21}$$
Let $\alpha_{\text{eff}} = \kappa e^{L_{\mathcal{F}}\Delta t_{\text{corr}}}$. Let $B = \kappa\mathcal{E}_{\text{ODE}} + \delta_{\mathcal{C}}$. Then the recurrence relation for the error after correction is:
$$e_{k+1}^+ \le \alpha_{\text{eff}} e_k^+ + B. \tag{22}$$
This is a linear first-order non-homogeneous recurrence relation. Unrolling this for $K$ steps, starting from $e_0^+$:
$$\begin{aligned}
e_K^+ &\le \alpha_{\text{eff}} e_{K-1}^+ + B \\
&\le \alpha_{\text{eff}}(\alpha_{\text{eff}} e_{K-2}^+ + B) + B = \alpha_{\text{eff}}^2 e_{K-2}^+ + \alpha_{\text{eff}} B + B \\
&\cdots \\
&\le \alpha_{\text{eff}}^K e_0^+ + B \sum_{j=0}^{K-1} \alpha_{\text{eff}}^j.
\end{aligned} \tag{23}$$
Since $\alpha_{\text{eff}} < 1$, the geometric series sum is $\sum_{j=0}^{K-1} \alpha_{\text{eff}}^j = \frac{1-\alpha_{\text{eff}}^K}{1-\alpha_{\text{eff}}}$. So,
$$e_K^+ \le \alpha_{\text{eff}}^K e_0^+ + B\frac{1 - \alpha_{\text{eff}}^K}{1 - \alpha_{\text{eff}}}. \tag{24}$$
This can be rewritten as:
$$e_K^+ \le \alpha_{\text{eff}}^K e_0^+ + \frac{B}{1-\alpha_{\text{eff}}} - \frac{B\alpha_{\text{eff}}^K}{1-\alpha_{\text{eff}}} = \alpha_{\text{eff}}^K \left(e_0^+ - \frac{B}{1-\alpha_{\text{eff}}}\right) + \frac{B}{1-\alpha_{\text{eff}}}. \tag{25}$$
The constant $C_2$ defined in the theorem is $C_2 = \frac{\kappa\mathcal{E}_{\text{ODE}}+\delta_{\mathcal{C}}}{1-\alpha_{\text{eff}}} = \frac{B}{1-\alpha_{\text{eff}}}$. This is the asymptotic error floor for $e_K^+$. Substituting $C_2$ into $Eq(7)$, we get:
$$e_K^+ \le \alpha_{\text{eff}}^K \left(e_0^+ - C_2\right) + C_2. \tag{26}$$
This matches the form of Equation (15) in the theorem statement, if we interpret $\|e(t)\|$ as $e_K^+$ (error after $K$ corrections) and $e(t_0)$ as $e_0^+$ (error after the initial, possibly hypothetical, correction, which serves as the starting point for this recurrence). The number of correction steps is $K = \lfloor(t - t_0)/\Delta t_{\text{corr}}\rfloor$.

Now for:
$$\|e(t)\| \le C_1 \cdot \alpha_{\text{eff}}^{\lfloor(t-t_0)/\Delta t_{\text{corr}}\rfloor} + C_2. \tag{27}$$
From $e_K^+ \le \alpha_{\text{eff}}^K(e_0^+ - C_2) + C_2$:

 * If $e_0^+ - C_2 \ge 0$, then we can set $C_1 = e_0^+ - C_2$. The bound becomes $C_1\alpha_{\text{eff}}^K + C_2$.
 * If $e_0^+ - C_2 < 0$, then $e_0^+ < C_2$. Since $\alpha_{\text{eff}}^K > 0$, the term $\alpha_{\text{eff}}^K(e_0^+ - C_2)$ is negative.

Thus, $e_K^+ \le C_2 - \alpha_{\text{eff}}^K(C_2 - e_0^+) < C_2$. In this case, setting $C_1 = 0$ ensures $e_K^+ \le 0 \cdot \alpha_{\text{eff}}^K + C_2 = C_2$, which is true. So, $C_1 = \max(0, e_0^+ - C_2)$ correctly captures both cases and matches the definition in the theorem. The bound demonstrates that if $\alpha_{\text{eff}} < 1$, the term $\alpha_{\text{eff}}^K \to 0$ as $K \to \infty$. Consequently, the error $e_K^+$ converges towards the asymptotic error floor $C_2$.

# G  External Results

## G.1  More Visualization Cases.

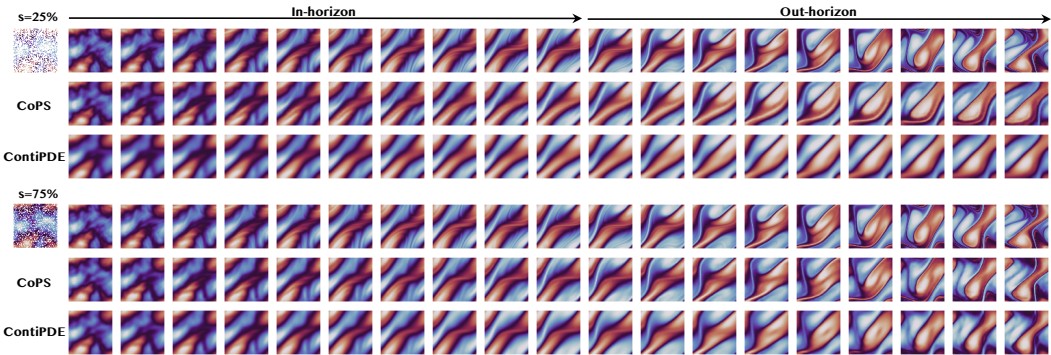

Figure 5: Qualitative comparison of our method and ContiPDE on the Navier-Stokes dataset under sparse initial observations (25% and 75%). In each panel, the first row displays the ground truth evolution, spanning both in-horizon (training) and out-horizon (extrapolation) timesteps. The second and third rows depict the predictions generated by our proposed CoPS and ContiPDE.

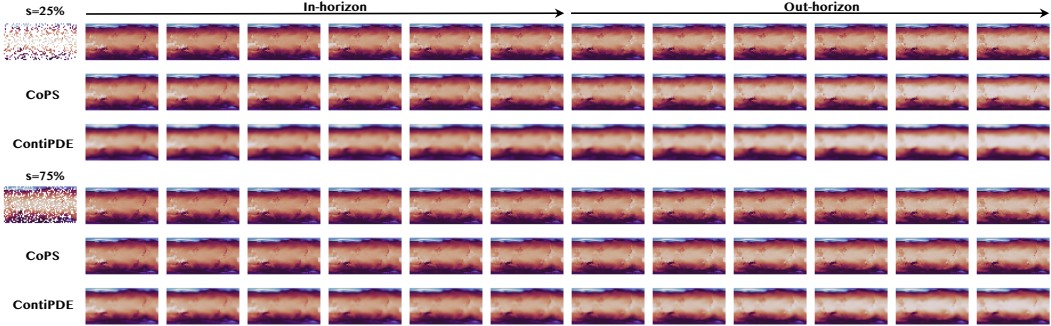

Figure 6: Qualitative comparison of our method and ContiPDE on the WeatherBench dataset under sparse initial observations (25% and 75%). In each panel, the first row displays the ground truth evolution, spanning both in-horizon (training) and out-horizon (extrapolation) timesteps. The second and third rows depict the predictions generated by our proposed CoPS and ContiPDE.

To more clearly demonstrate the performance of our method, we present detailed visualizations here. Figure 5 and Figure 6 provide qualitative illustrations of CoPS's capabilities in modeling complex fluid dynamics from sparse initial data on the Navier-Stokes dataset and the WeatherBench dataset, comparing its performance against ground truth and the ContiPDE baseline. We examine scenarios with initial conditions derived from both 25% and 75% of the full observations. The ground truth (first row in each panel) clearly shows the evolution of intricate flow patterns, including vortex formation and propagation, across both in-horizon and out-of-horizon time steps. When initialized with only 25% observations (top panel), our CoPS model (second row) successfully captures the essential dynamics, maintaining structural integrity and accurately predicting the advection of vortices well into the extrapolation phase. In contrast, ContiPDE (third row), while capturing the general flow, struggles more with the sparsity, leading to predictions that are smoother and lose some of the high-frequency details present in the ground truth, particularly in the out-of-horizon predictions. This suggests CoPS's encoding mechanism and the interplay between its continuous ODE evolution and discrete auto-correction are more effective in inferring the complete state from limited information and robustly propagating it. Even with 75% initial data (bottom panel), where both models perform better, CoPS consistently exhibits a closer match to the ground truth's finer details and long-term behavior, highlighting its enhanced capacity for accurate and stable long-range forecasting in continuous spatio-temporal domains.

## G.2 Hyperparameter Analysis

We conduct sensitivity experiments with regard to correction hyperparameter ($\lambda$) on Navier-Stokes and Prometheus datasets. To resolve your concern, we conduct experiments on both In-t and Ext-t settings with observation subsampling ratio of 50%. The results on Ext-t setting demonstrate the long-term prediction performance. The results are shown in Table 4, which indicate that neural auto-correction can indeed improve the performance of our method, and the experimental results are robust to hyperparameter $\lambda$.

Table 4: Hyperparameter sensitivity of $\lambda$ on the Navier-Stokes and Prometheus datasets with 50% subsampling ratio. We report MSE for In-t and Ext-t settings.

|  | $\lambda = 0$ | $\lambda = 0.1$ | $\lambda = 0.2$ | $\lambda = 0.5$ | $\lambda = 1.0$ |
|---|---|---|---|---|---|
| NAVIER-STOKES (IN-T) | 3.244E-03 | 3.017E-03 | 2.925E-03 | 2.832E-03 | 2.964E-03 |
| NAVIER-STOKES (EXT-T) | 6.635E-03 | 6.172E-03 | 5.873E-03 | 5.764E-03 | 5.828E-03 |
| PROMETHEUS (IN-T) | 3.623E-03 | 3.542E-03 | 3.495E-03 | 3.374E-03 | 3.545E-03 |
| PROMETHEUS (EXT-T) | 7.016E-03 | 6.823E-03 | 6.747E-03 | 6.678E-03 | 6.837E-03 |

## G.3 Noise Disturbance Robustness.

To evaluate the robustness of our model, we present the effects of observational noise on its performance and compare these results with those of other models. We quantify the noise using the channel-specific standard deviation tailored to the dataset and have trained the model under various noise intensities (noise ratios set at 1%, 5%, 10%, 15%, and 20%). Experiments are conducted on and Navier-Stokes and Prometheus datasets. Observations from Figure 7 reveal that the proposed `CoPS` effectively maintains its performance with noise ratio below 5%, and demonstrates significant advantages over other baseline models when noise ratio increases over 10%. In contrast, we observe a more pronounced performance degradation in the two interpolation-based baseline models as noise ratio raises over 10%, which indicates their weaker robustness against noise interference.

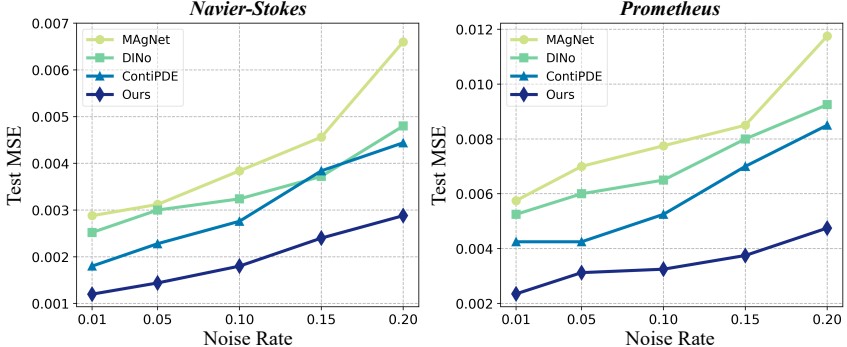

Figure 7: Performance with regard to noise disturbance.

