# OpenReview forum: "Breaking the Discretization Barrier of Continuous Physics Simulation Learning"
_NeurIPS.cc/2025/Conference — NeurIPS 2025 poster_

### Official Review · Reviewer_doai · 2025-06-10

**Clarity:** 2
**Significance:** 2
**Originality:** 2
**Rating:** 4
**Confidence:** 3

**Summary:**

This paper introduces CoPS, a method for modeling physics simulations from partial spatial observations. The authors employ Gabor filters to generate representations and utilize Multiscale Neural Graph ODEs to model the dynamics. Additionally, they propose a Markovian correction term to mitigate cumulative errors in the ODE process. Experiments across various dynamical systems demonstrate improved performance in sparse observation, super-resolution, and temporal interpolation and extrapolation tasks.

**Questions:**

- What is the rationale behind using Gabor filters?
- How is extrapolation/interpolation performed for arbitrary times when the correction is designed for discrete steps? Is the time interval $t_{k+1} - t_k$ always constant?
- What is the effect of the correction hyperparameter?
- Is Mean Squared Error (MSE) the most suitable metric for evaluating the impact of various components, or should other metrics like SSIM or PSNR be considered?

**Ethical Concerns:**

["NO or VERY MINOR ethics concerns only"]

**Final Justification:**

The authors' rebuttal clarified most of the questions I had about this work. Although, I still maintain that the work has limited novelty -- based on the results and discussion, I would recommend Weak Acceptance by increasing by score from 3 to 4.

**Paper Formatting Concerns:**

Marcov -> Markov in the abstract

**Quality:**

3

**Strengths And Weaknesses:**

Strengths
- The model effectively handles multiple tasks, including interpolation, extrapolation, and super-resolution, with good results.
- The authors provide theoretical justifications for the impact of latent corrections.
- Despite its complexity, the paper is well-structured and easy to follow.

Weakness:
- While the work integrates numerous components for optimal results, many of these components are derived from previous studies, limiting the novelty of the approach.
- The assumption that all dynamics can be learned from a single initial observation at time $t_0$ is questionable.
- The phrase "L arbitrary consecutive time points" feels like an oxymoron.

---

> ### Author Rebuttal · Authors · 2025-07-31
>
> Dear Reviewer doai,
>
> Thank you for your valuable feedback on our manuscript. We have taken your comments seriously and have made the necessary revisions and additions to address the concerns raised. Below is our point-by-point rebuttal:
>
> > Q1. While the work integrates numerous components for optimal results, many of these components are derived from previous studies, limiting the novelty of the approach.
>
> A1. Thanks for your valuable comment. We agree that some of our core components are built upon powerful existing methods, and we have made sure to cite these foundational works. However, we wish to clarify that the novelty of our method lies not in its individual parts, but in the complete and functional system they form when synergistically integrated.
>
> Specifically, we first employ a Gabor filter-based multiplicative filter network (MFN) to encode coordinate information into the initial state as a global frequency representation. Next, to address the challenge that grids in continuous modeling can be arbitrary and dynamic, we introduce a customized regular grid and design a dedicated message-passing mechanism to map features from the original unstructured domain onto it.
>
> Subsequently, for latent dynamics modeling on this regularized grid, we introduce the multi-scale graph ODE (MGO). While its hierarchical structure is inspired by GraphCast, we integrate it with an attention-based, multi-level message-passing scheme and uniquely combine it with a Neural ODE to learn both global and local dynamics in a continuous-time fashion. Furthermore, to mitigate the well-known issue of error accumulation in Neural ODEs, which arises from the difficulty of learning complex nonlinear features, we propose our neural auto-correction (NAC) module. This component performs adaptive corrections at discrete time steps.
>
> The synergy of these modules is paramount to our model's success. Our extensive ablation study (Table 3) empirically validates this claim, showing that removing any of these components results in a significant degradation in performance. We will ensure this is articulated more clearly in the revised manuscript.
>
> > Q2. The assumption that all dynamics can be learned from a single initial observation at time $t_0$ is questionable.
>
> A2. Thanks for your comment. Our goal is not to learn from a single, isolated training example. Instead, our model is trained on a large dataset of **trajectories**, where each trajectory provides numerous examples of an initial state and its corresponding future states (e.g., pairs of $(u(t_0), u(t_1))$, $(u(t_0), u(t_2))$, etc.). By learning from thousands of such examples, the model's objective is to approximate a universal **evolution operator**, Φ, that governs how the system evolves from any valid initial state $u(t_0)$ to a future state $u(t_k)$. Therefore, at **inference time**, using a single observation at $t_0$ is the standard procedure to test the generalization and accuracy of the learned operator Φ.
>
> This "Initial Value Problem" setup is not a novel invention of ours; rather, it is in direct alignment with the well-established paradigm of traditional numerical simulation that has been used for decades. There exist several Influential works like GraphCast [1] and Pangu-Weather [2]. The tremendous success of these models provides compelling evidence that for many high-dimensional physical systems, the state at a single point in time contains sufficient information to determine its short-to-medium-term evolution. Our work proudly follows this validated research framework.
>
> > Q3. The phrase "L arbitrary consecutive time points" feels like an oxymoron.
>
> A3. Thanks for your valuable feedback. To be precise: during the training stage, our model is indeed supervised using observations at a sequence of discrete time points from the dataset. The term 'arbitrary' was intended to describe the powerful capability of our model at the inference and evaluation stage. We recognize this distinction is critical and will state it with much greater precision in the revised manuscript to eliminate any ambiguity.
>
>
> > Q4. What is the rationale behind using Gabor filters?
>
> A4. Thank you for your insightful comment. The fundamental rationale for using Gabor filters is to overcome the well-documented spectral bias of standard MLP networks, which struggle to represent high-frequency functions and thus tend to produce overly smooth reconstructions of complex physical fields. To address this, our design employs Gabor filters to transform raw positional coordinates into a feature representation rich with multi-scale frequency and orientation information, directly empowering the network to learn and reconstruct the high-frequency details essential for physical fidelity.
>
> > Q5. How is extrapolation/interpolation performed for arbitrary times when the correction is designed for discrete steps? Is the time interval $t_{k+1} - t_{k}$ always constant?
>
> A5. Thanks for your comment. The task of predicting the system's state at any arbitrary time, whether for interpolation or extrapolation, is handled by our multi-scale Graph ODE module. It acts as the continuous dynamics engine. To predict a state at a query time $t_{query}$, we start from the most recent corrected state at a discrete time step $t_k$ and integrate the learned ODE forward from $t_k$ to $t_{query}$. This integration process is, by definition, continuous. The neural auto-correction (NAC) module, by contrast, acts as a stabilizer. It is invoked only at pre-defined, discrete time steps ($t_k, t_{k+1}$, etc.) to "project" the ODE's state back onto a learned manifold of physically plausible solutions, thereby preventing the long-term accumulation of numerical error. It does not participate in the prediction at arbitrary times between these steps.
>
> Then，for the time interval $t_{k+1} - t_{k}$, we did indeed use a constant correction interval, which typically corresponded to the interval between two consecutive frames in the training data (e.g., $\Delta t = 1.0$).
>
> > Q6. What is the effect of the correction hyperparameter?
>
> A6. Thanks for your comment. We conduct sensitivity experiments w.r.t correction hyperparameter (λ) on Navier-Stokes and Prometheus datasets. To resolve your concern, we conduct experiments on both In-t and Ext-t settings with observation subsampling ratio of 50%. The results on Ext-t setting demonstrate the long-term prediction performance. The results are shown below, which indicate that neural auto-correction can indeed improve the performance of our method, and the experimental results are robust to hyperparameter λ.
>
> | | λ=0|λ=0.1|λ=0.2|λ=0.5|λ=1.0|
> | - | - | - | - | - | - |
> |  Navier-Stokes(In-t)|3.244E-03|3.017E-03|2.925E-03|2.832E-03| 2.964E-03|
> |  Navier-Stokes(Ext-t)|6.635E-03|6.172E-03|5.873E-03 | 5.764E-03| 5.828E-03|
> |  Prometheus(In-t)|3.623E-03|3.542E-03 | 3.495E-03 | 3.374E-03| 3.545E-03 |
> |  Prometheus(Ext-t)|7.016E-03|6.823E-03|6.747E-03|6.678E-03| 6.837E-03 |
>
> > Q7. Is Mean Squared Error (MSE) the most suitable metric for evaluating the impact of various components, or should other metrics like SSIM or PSNR be considered?
>
> A7. Thanks for your valuable comment. Our primary rationale for selecting Mean Squared Error (MSE) is its direct relevance to physical fidelity. In many scientific and engineering domains, MSE (or its square root, RMSE) is the standard and most direct metric for quantifying point-wise accuracy between a prediction and the ground truth, which aligns perfectly with our core objective of precise prediction.
>
> However, we fully agree that metrics like the SSIM and PSNR are indeed excellent for evaluating the perceptual and structural qualities. To provide a more holistic evaluation and address the concern, we have conducted experiments on the Navier-Stokes dataset (using 50% subsampling for training) to report these additional metrics. The results are summarized below.
>
> | Model | Task | MSE (↓) | SSIM (↑) | PSNR (↑) |
> | :--- | :--- | :--- | :--- | :--- |
> | MAgNet | In-t | 2.60E-02 | 0.826 | 25.85 |
> | | Ext-t | 4.29E-02 | 0.743 | 23.68 |
> | DINo | In-t | 1.07E-02 | 0.892 | 29.69 |
> | | Ext-t | 1.76E-02 | 0.834 | 27.54 |
> | ContiPDE | In-t | 8.34E-03 | 0.902 | 30.79 |
> | | Ext-t | 1.29E-02 | 0.873 | 28.89 |
> | **Ours (CoPS)** | **In-t** | **5.76E-03** | **0.942** | **33.40** |
> | | **Ext-t** | **9.82E-03** | **0.921** | **31.08** |
>
> As the table demonstrates, our proposed method, CoPS, not only achieves superior performance on the MSE metric but also consistently outperforms all baseline models on both SSIM and PSNR. This confirms that our approach excels not only in point-wise accuracy but also in faithfully reconstructing the structural and fine-grained details of the underlying physical field. We will add this experiment in our revised manuscript.
>
> > Q8. Paper formatting concerns.
>
> A8. Thanks for your comment. We have performed a thorough proofread of the entire manuscript, correcting specific typographical errors (such as the instance of "Marcov" to "Markov") and making broader revisions to improve overall consistency, clarity, and presentation.
>
> ---
> [1] Lam R, et al. Learning skillful medium-range global weather forecasting. Science, 2023.
>
> [2] Bi K, et al. Accurate medium-range global weather forecasting with 3D neural networks. Nature, 2023.

---

> > ### Comment · Reviewer_doai · 2025-08-02
> > **Thank you for the clarification**
> >
> > Dear Authors,
> >
> > Thank you for the additional results and clarification. Most of my questions have been clarified -- however, I am still confused about Q2. How do you predict a dynamical system from just the initial state? For example: wouldn't you need atleast 2 time-steps to measure velocity or am I missing something?

---

> > > ### Author Response · Authors · 2025-08-03
> > > **Thanks for your response!**
> > >
> > > We are pleased to know that most of your concerns have been addressed! And we appreciate the opportunity to respond to your insightful question regarding Q2. Your confusion is entirely understandable, as it highlights a critical distinction between how our model is trained and how it is used for prediction. The core of our response is this: your intuition that at least two time-steps are needed to infer dynamic properties like velocity is correct, and this requirement is fulfilled during our model's training phase. However, once the model has learned the system's dynamics and its parameters are frozen, it can then perform extrapolation from a single initial state during the inference phase.
> > >
> > > A dynamical system is fundamentally defined by a set of rules that govern its evolution over time. For a given state $z$ at time $t$, these rules determine the state at a future time, $t+\Delta t$. The primary objective of our deep learning model is to learn a function, $f_{\theta}$, that approximates this mapping, $z_{t+\Delta t} = f_{\theta}(z_t)$, thereby capturing the underlying rules of the system without needing them to be explicitly programmed.
> > >
> > > During the **training phase**, our model operates under a supervised learning paradigm. We provide it with a vast number of sequences from the dynamical system, specifically pairs of states $(z_t, z_{t+\Delta t})$. The model uses $z_t$ as input to make a prediction, and its internal parameters ($\theta$) are optimized to minimize the discrepancy between its prediction and the true future state. This allows the model  to implicitly learn the system's velocity, acceleration, and other temporal evolution characteristics. To facilitate training, future states are usually expressed in discrete time steps.
> > >
> > > Conversely, during the **inference phase**, the model's parameters, $\theta$, are fixed. The model now embodies the learned dynamics function, $f_{\theta}$. At this stage, we only need to provide a single initial state, $z_0$. The model applies its learned function to predict the next state: $z_1 = f_{\theta}(z_0)$. This prediction, $z_1$, is then used as the new input to predict the subsequent state, $z_2 = f_{\theta}(z_1)$. This autoregressive process can be repeated to generate an entire future trajectory from just the initial condition.
> > >
> > > This methodology, where training requires sequence data but inference can be initiated from a single state, is a well-established and powerful paradigm in modern data-driven forecasting, like Pangu-Weather [1], Aardvark [2], Oneforecast [3], and Triton[4].
> > >
> > > Our method builds upon this foundation and takes it a significant step further. While the aforementioned approaches learn a mapping between discrete ($t$ and $t+\Delta t$), our framework embeds a Graph ODE to learn a continuous vector field that governs the system's dynamics, represented by the differential equation $\frac{dz}{dt} = f_{\theta}(z, t)$. Consequently, this enables two key advantages over discrete-time models: (1) we can supervise the training using states at any arbitrary, non-discrete moments in time, and (2) during inference, we can query the model for a future state at any continuous time point, not just at fixed intervals.
> > >
> > > ---
> > > [1] Bi K, et al. Accurate medium-range global weather forecasting with 3D neural networks. Nature, 2023.
> > >
> > > [2] Allen A, et al. End-to-end data-driven weather prediction. Nature, 2025.
> > >
> > > [3] Gao Y, et al. OneForecast: A Universal Framework for Global and Regional Weather Forecasting. ICML, 2025.
> > >
> > > [4] Wu H, et al. Advanced long-term earth system forecasting by learning the small-scale nature. Arxiv, 2025.
> > >
> > > ---
> > >
> > > Thanks again for your valuable feedback! Please kindly let us know if you have any questions further!
> > >
> > > Warm regards,
> > >
> > > the Authors

---

> > > > ### Comment · Reviewer_doai · 2025-08-05
> > > >
> > > > Could you please clarify how the train and test sets are different?
> > > > To be precise - In the Navier-Stokes example: For the training phase do you keep the coefficients of the PDE fixed and generate data for a variety of initial conditions or do you generate data by changing the coefficients of the PDE?
> > > >
> > > > Similarly, How is the test data generated?

---

> > > > > ### Author Response · Authors · 2025-08-06
> > > > > **Thanks for your response!**
> > > > >
> > > > > Dear Reviewer doai,
> > > > >
> > > > > Thank you for your insightful question. For the simulated **Navier-Stokes** dataset, the train and test sets differ only by their **initial conditions**. All simulations were generated from the Navier-Stokes equation with a constant Reynolds number of 1e-5. We utilized a total of 1200 independent simulation samples. Each sample was initiated with a unique, random velocity field and simulated for the same number of timesteps. These 1200 samples were then partitioned in a 7:2:1 ratio into training, validation, and test sets.
> > > > >
> > > > > For real-world data like **WeatherBench**, we use the historical ERA5 global atmospheric reanalysis data. The data was partitioned strictly by date to prevent any data leakage from the future into the training process. Specifically, the training set uses data from 1979-2018, the validation set from 2019, and the test set from 2020-2022.
> > > > >
> > > > > We will incorporate a detailed description of the data partitioning strategy into the revised version for clarity.
> > > > >
> > > > > Thanks again for your valuable feedback! Please kindly let us know if you have any questions further!
> > > > >
> > > > > Sincerely,
> > > > >
> > > > > the Authors

---

> > > > > > ### Comment · Reviewer_doai · 2025-08-06
> > > > > >
> > > > > > Thank you, authors. I think it's important to clarify these details in the paper, so I'll look forward to seeing them included in the manuscript. I have increased my score to a 4.

---

> > > > > > > ### Author Response · Authors · 2025-08-06
> > > > > > > **Thanks for your feedback and support!**
> > > > > > >
> > > > > > > Dear Reviewer doai,
> > > > > > >
> > > > > > > We sincerely appreciate your valuable feedback and recognition! We will definitely incorporate your suggestions into our revised version. Please kindly let us know if you have any questions further!
> > > > > > >
> > > > > > > Best regards,
> > > > > > >
> > > > > > > the Authors

---

### Official Review · Reviewer_RVtr · 2025-06-30

**Clarity:** 3
**Significance:** 3
**Originality:** 2
**Rating:** 5
**Confidence:** 2

**Summary:**

The authors propose a data-driven framework CoPS for the continuous modelling and prediction of dynamical systems. This allows for evolving systems in a grid-free and timestep-free way, constrained by sparse observations in time and space. The authors do this by combining several methodologies, namely multiplicative filter network for encoding, message passing for spatial modelling, and multi-scale graph ODEs for temporal modelling (in latent space). They show that CoPS outperforms a variety of benchmark on many continuous modelling for synthetic and real-world datasets.

**Questions:**

CoPS first performs an encoding to a customized but concrete grid. From the text in Sec. 3.1, it seems that points are mapped onto a uniform grid structure in the initial encoding stage. If this is the case, and there are sparse observations where points are sometimes very close together and sometimes very far apart, then this would require a large but very fine-grained uniform grid, leading to many many grid cells and thus be very expensive, and likely infeasible in a 3D setting. Could the authors please clarify?

Could the authors provide more details on the baselines evaluation? Are the parameter counts for all the baselines roughly the same? How many graph scales are used for CoPS for the different tasks? What is the inference time for this method as compared to the baselines?

**Ethical Concerns:**

["NO or VERY MINOR ethics concerns only"]

**Final Justification:**

For both of my main concerns, the authors provided sufficient clarification. They also ran additional experiments and resolved my concerns about memory usage and scalability of the method.

**Limitations:**

yes

**Quality:**

3

**Strengths And Weaknesses:**

The authors address discretization in both spatial and temporal dimensions, which allows the framework to be highly flexible.

The framework builds a collection of graphs on multiple scales, performs processing on each graph/scale independently, then mixes information between scales using an attention mechanism.

The authors use a neural correction module to address limitations when using numerical solvers. The mix of the standard solver to perform most of the work and the neural corrector to only model the residual is very nice.

---

> ### Author Rebuttal · Authors · 2025-07-31
>
> Dear Reviewer RVtr,
>
> We sincerely appreciate the time you’ve dedicated to reviewing our paper, as well as your valuable insights and support. Your positive feedback is highly motivating for us. Below, we address your primary concern and offer further clarification.
>
> > Q1. CoPS first performs an encoding to a customized but concrete grid. From the text in Sec. 3.1, it seems that points are mapped onto a uniform grid structure in the initial encoding stage. If this is the case, and there are sparse observations where points are sometimes very close together and sometimes very far apart, then this would require a large but very fine-grained uniform grid, leading to many many grid cells and thus be very expensive, and likely infeasible in a 3D setting. Could the authors please clarify?
>
> A1. Thanks for your valuable comment. This concern is entirely valid under the assumption that the grid resolution must be directly tied to the input data's spatial density. However, our approach is specifically designed to decouple the resolution of our customized latent grid from the spatial distribution of the sparse input observations, thereby avoiding this very issue.
>
> First, to clarify the mechanism: the customized grid in our framework acts as a structured, latent manifold whose resolution is a fixed hyperparameter, not a function of input point proximity. An observation point $p_i$ is mapped only to the vertices of the single grid cell that contains it. Even if ten points are tightly clustered within one cell, they all connect to the same few vertices. Information is then aggregated onto these vertices via a learnable message-passing scheme, which uses relative position embeddings to preserve the sub-cell spatial information. This design ensures that the primary computational cost is proportional to the chosen grid size (e.g., $H \times W$), which remains constant and predictable, regardless of input point distribution.
>
> To empirically validate this principle and directly address the insightful concern about 3D feasibility, we conducted new experiments on the 3D WeatherBench2 dataset [1]. We evaluated our model's performance and resource consumption across low, medium, and high grid resolutions. The results are presented below.
>
> |  | Low Res (32x64x60) | Medium Res (90x180x60) | High Res (180x360x60) |
> | :--- | :--- | :--- | :--- |
> | Total Grid Points | 123 K | 972 K | 3.89 M |
> | GPU Memory (VRAM) | 3.5 GB | 8.9 GB | 20.5 GB |
> | Inference Time (s) | 4.5 s | 19.0 s | 58.0 s |
>
> The results reveals two crucial findings:
> 1.  **Memory usage scales gracefully.** While the number of grid points grows by over 30 times from low to high resolution, the required VRAM increases by less than 6 times. This sub-linear growth is far from the prohibitive explosion one might fear and demonstrates the excellent memory efficiency of our approach in high-dimensional settings.
> 2.  **Inference time shows strong sub-linear scaling.** Similarly, the inference time does not exhibit exponential growth but scales in a highly favorable, near-linear fashion relative to the number of grid points. This confirms that the computational cost is dictated by our manageable latent grid, not by complex interactions tied to input data geometry.
>
> In all, both our model's design principle and these new 3D empirical results demonstrate that our method is indeed efficient and feasible for large-scale applications. The decoupling of the latent grid from the input data distribution is the key mechanism that ensures this scalability. We will include this experiment in our revised version.
>
>
> A2. Thanks for your valuable feedback. For the experiments presented in the paper, our CoPS model was configured with a consistent set of key hyperparameters to ensure robustness. A more detailed list of all model hyperparameters is available in the Appendix D.1 and D.3. Specifically, in the multi-scale Graph ODE (MGO) module, since the spatial resolutions of our experimental datasets are of a comparable order of magnitude, we adopted a consistent configuration of four graph scales across all our experiments.
>
> Then, to evaluate the efficiency our method and baselines, we have conducted experiments on the Navier-Stokes dataset. The results, presented in the table below, demonstrate that our method ranks second in raw efficiency metrics behind MAgNet, which employs nearest neighbor interpolation technique to generalize to new query points. The results clearly show that CoPS achieves its state-of-the-art predictive performance (as detailed in the paper)) with exceptional efficiency. This favorable balance underscores the effectiveness and practicality of our proposed architecture. We will incorporate this table into our revised manuscript.
>
> |   |Param| Training time | Inference time |
> | ------------ | ------ | ------   | ------ |
> | MAgNet    | 14.25 MB  | 4.11 h   | 3.25 s |
> | DINo      | 25.17 MB  | 6.74 h   | 6.38 s |
> | ContiPDE  | 32.48 MB  | 8.92 h   | 7.56 s |
> | Ours      | 22.53 MB  | 5.27 h   | 5.51 s |
>
> ---
> [1] Rasp S, et al. WeatherBench 2: A benchmark for the next generation of data‐driven global weather models. Journal of Advances in Modeling Earth Systems 2024.

---

> > ### Comment · Reviewer_RVtr · 2025-08-04
> >
> > Thank you for the response and for running the additional experiments. I now have a better understanding of the motivation and design for the framework proposed in this work. My concerns have been addressed, especially by the new 3D experiment. I have updated my score accordingly. Good luck.

---

### Official Review · Reviewer_vtdp · 2025-07-01

**Clarity:** 2
**Significance:** 3
**Originality:** 2
**Rating:** 4
**Confidence:** 4

**Summary:**

This study introduces CoPS, a data-driven framework designed to model continuous spatiotemporal dynamics in complex physical systems using partial observations. CoPS overcomes the discretization constraints of traditional methods by merging a multiplicative filter network to encode spatial coordinates with observation data, and multi-scale graph-based ordinary differential equations (ODEs) paired with a Markovian neural auto-correction module. Through experiments on diverse datasets—including Navier-Stokes fluid simulations and weather forecasting—the framework demonstrates enhanced performance in continuous space-time modeling and long-term prediction, particularly under sparse data conditions.

**Questions:**

Please provide responses to the above listed weakness.

Also, there are following additional questions:

1. Should provide evaluations on efficiency of the proposed method and baselines, also the comparison of the number of parameters.
2. How to decide the values of $\mu$ and $\gamma$ in Eq. (2)?
3. How is relative position embedding designed?
4. What is the effect of the hyperparameter $\lambda$?

**Ethical Concerns:**

["NO or VERY MINOR ethics concerns only"]

**Final Justification:**

My concerns are well clarified and addressed. I think the paper is now technically solid with comprehensive evaluation.
Thus, I raise my rating to "4: borderline accept". The reason why I do not provide a "5: accept" score for this paper is that I think its novelty is not strong enough.

**Limitations:**

Yes

**Quality:**

2

**Strengths And Weaknesses:**

Strengths:

1.  CoPS proposes a data-driven approach combining multiplicative filter networks, customized geometric grids, and multi-scale graph ODEs to overcome discretization limitations in physics simulation, enabling spatio-temporal continuous prediction from partial observations. The framework integrates message-passing mechanisms, multi-scale graph ODEs for dynamic modeling, and a Markov-based neural auto-correction module for robust long-term predictions and error correction in nonlinear systems.

2. Experiments on diverse datasets demonstrate state-of-the-art performance, especially in sparse data scenarios (25% observations) and long-term extrapolation beyond training horizons.

Weaknesses:

1. Core components (multiplicative filter network, multi-scale graph structure) resemble existing methods (MFN, GraphCast), lacking fundamental component-level innovations and relying on modular combinations.

2. The inference cost is high since it involves an ODE simulation for a query point, which makes it expensive.

3. The presentation of this paper is poor. In particular,  the Methodology Section is hard to follow. Some points need to be clarified:

    (1) The structure and notations of the Methodology Section should be improved. For example, it is unclear how many models should be learned and how they are trained in this section.


    (2). What is the explicit formulation of the ODE in Eq. (7)? Why the continuous-time evolution could be modeled as such an ODE involving hierarchical graph structure G and hidden representations h?

    (3). The mechanism of Neural Auto-correction is not very clear. Why it could help to reduce error accumulation? What is the relation between the module $r_\phi$ and $E(\cdot)$ and $D(\cdot)$?

---

> ### Author Rebuttal · Authors · 2025-07-31
>
> Dear Reviewer vtdp,
>
> We sincerely appreciate the time you’ve dedicated to reviewing our paper, as well as your valuable insights and support. Your positive feedback is highly motivating for us. Below, we address your primary concern and offer further clarification.
>
> > Q1. Core components (multiplicative filter network, multi-scale graph structure) resemble existing methods (MFN, GraphCast), lacking fundamental component-level innovations and relying on modular combinations.
>
> A1. Thanks for your valuable feedback. We agree that some of our core components are built upon powerful existing methods, and we have made sure to cite these foundational works. However, we wish to clarify that the novelty of our method lies not in its individual parts, but in the complete and functional system they form when synergistically integrated.
>
> Specifically, we first employ a Gabor filter-based multiplicative filter network (MFN) to encode coordinate information into the initial state as a global frequency representation. Next, to address the challenge that grids in continuous modeling can be arbitrary and dynamic, we introduce a customized regular grid and design a dedicated message-passing mechanism to map features from the original unstructured domain onto it.
>
> Subsequently, for latent dynamics modeling on this regularized grid, we introduce the multi-scale graph ODE (MGO). While its hierarchical structure is inspired by GraphCast, we integrate it with an attention-based, multi-level message-passing scheme and uniquely combine it with a Neural ODE to learn both global and local dynamics in a continuous-time fashion. Furthermore, to mitigate the well-known issue of error accumulation in Neural ODEs, which arises from the difficulty of learning complex nonlinear features, we propose our neural auto-correction (NAC) module. This component performs adaptive corrections at discrete time steps.
>
> The synergy of these modules is paramount to our model's success. Our extensive ablation study (Table 3) empirically validates this claim, showing that removing any of these components results in a significant degradation in performance. We will ensure this is articulated more clearly in the revised manuscript.
>
> > Q2. The inference cost is high since it involves an ODE simulation for a query point, which makes it expensive.
>
> A2. Thanks for your valuable comment regarding inference cost. We would like to clarify that the computationally intensive ODE simulation is performed only once for the entire system's latent state on a regularized grid, not on a per-query-point basis. Specifically, our method first performs a one-time encoding of the initial $N$ sensor observations onto a grid of size $G$ (complexity $\mathcal{O}(N+G)$). The multi-scale Graph ODE then evolves the state of this entire grid over $S$ integration steps, which constitutes the primary computational cost of $\mathcal{O}(S \cdot G)$. Crucially, this cost is independent of the number of final query points, $M$. Finally, decoding the value at each of the $M$ query locations is a highly efficient decoder with a complexity of $\mathcal{O}(M)$. Therefore, the total inference cost is $\mathcal{O}(N+G + S \cdot G + M)$, demonstrating that our framework amortizes the expensive dynamics simulation over an arbitrary number of spatial queries, with the cost scaling with temporal granularity ($S$) rather than spatial query density ($M$).
>
> > Q3. The structure and notations of the Methodology Section should be improved. For example, it is unclear how many models should be learned and how they are trained in this section.
>
> A3. Thanks for your comment. We wish to clarify that our proposed CoPS is a single, end-to-end trainable architecture, where all components are optimized jointly. As we have detailed in our response to the first question (A1), our framework is a synergistic system composed of several key modules (MFN, MGO, and NAC) that collectively achieve our goal. We acknowledge that we will carefully revise the structure and notations throughout the Methodology section in the revised manuscript. Furthermore, we will add a new subsection at the end of this section to provide a clear summary of the model implementation.
>
> > Q4. What is the explicit formulation of the ODE in Eq. (7)? Why the continuous-time evolution could be modeled as such an ODE involving hierarchical graph structure G and hidden representations h?
>
> A4. Thanks for your comment. (Eq. 7) instantiates a multi-scale graph ODE. The core of our methodological contribution here is the synergistic integration where we embed our multi-scale graph structure, along with its corresponding message-passing mechanism, directly into the Neural ODE framework to parameterize the derivative function $\Phi$. This means that for any given state $h^t$, the function $\Phi$ computes its instantaneous rate of change, $\frac{dh^t}{dt}$, by performing a full multi-scale analysis as detailed in (Eq. 5) and (Eq. 6). Our rationale for this design is that many physical systems are governed by Partial Differential Equations (PDEs), where the temporal derivative of a field is determined by its current spatial configuration. The ODE framework provides a natural, continuous-time representation of this principle.
>
> > Q5. The mechanism of Neural Auto-correction is not very clear. Why it could help to reduce error accumulation? What is the relation between the module $r_\psi$ and $E(\cdot)$ and $D(\cdot)$?
>
> A5. Thanks for your comment. To clarify the notation, the neural auto-correction module $r_\psi$ in (Eq. 12) is indeed the entire composite function defined by the Encoder-Transition-Decoder structure, such that $r_\psi(\cdot) = \mathcal{D}(\mathcal{R}(E(\cdot)))$. The purpose of this Neural Auto-correction (NAC) module is to mitigate the problem of error accumulation inherent in long-term predictions with Neural ODEs. While the ODE provides a continuous evolution path, it relies on a single learned vector field, where small approximation errors can compound over time, causing the solution to drift into physically implausible regions of the state space. Our NAC module acts as a corrector at discrete time intervals. At each correction step, the encoder $E(\cdot)$ projects the state from the ODE into a more compact and stable latent space. Within this space, the transition block $\mathcal{R}(\cdot)$ applies a robust single-step mapping to learn latent correction. Finally, the decoder $D(\cdot)$ projects the corrected state back to the original feature space. In essence, this mechanism does not attempt to perfect the ODE's learned vector field itself, but rather periodically re-initializes and constrains the solution trajectory to a learned manifold of valid physical states.
>
> > Q6. Should provide evaluations on efficiency of the proposed method and baselines, also the comparison of the number of parameters.
>
> A6. Thanks for your comment. To evaluate the efficiency our method and baselines, we have conducted experiments on the Navier-Stokes dataset. The results, presented in the table below, demonstrate that our method ranks second in raw efficiency metrics behind MAgNet, which employs nearest neighbor interpolation technique to generalize to new query points. The results clearly show that CoPS achieves its state-of-the-art predictive performance with exceptional efficiency. We will incorporate this complete analysis into our revised manuscript.
>
> |   |Param (MB)| Training time (h) | Inference time (s) |
> | - | - | - | - |
> | MAgNet|14.25|4.11| 3.25|
> | DINo|25.17 |6.74| 6.38|
> | ContiPDE|32.48|8.92|7.56|
> | Ours|22.53|5.27|5.51 |
>
> > Q7. How to decide the values of $\mu$ and $\gamma$ in Eq. (2)?
>
> A7. Thanks for your comment. Within the Gabor filter, the values for $\mu$ and $\gamma$ in our model are determined through a two-stage process: a principled initialization followed by end-to-end optimization, as they are treated as learnable parameters. Specifically, $\mu$ is initialized uniformly at random within $[-1, 1)$ to ensure broad spatial coverage, while $\gamma$ is sampled from a Gamma distribution, providing a robust, non-negative prior for the filter scale. Crucially, both are then wrapped as trainable weights. This means their final values are not manually set but are automatically learned and fine-tuned by the optimizer via backpropagation during training. This design transforms positional information into a feature representation rich with multi-scale frequency and orientation information, empowering the network to learn and reconstruct high-frequency details within the physical field, which is fundamental for modeling spatial continuity. The detailed descriptions will be included in the revised version.
>
> > Q8. How is relative position embedding designed?
>
> A8. Thanks for your comment. The role of the relative position embedding in our message-passing scheme (Eq. 4) is to inject a geometric inductive bias into the feature propagation process. The relative position embedding $\phi(x_i, x_j)$ is generated by passing this difference vector through a single MLP, such that $\phi(x_i, x_j) = \text{MLP}(x_i - x_j)$. We will add this detail in the revised manuscript.
>
> > Q9. What is the effect of the hyperparameter λ?
>
> A9. Thanks for your comment. The effect of the hyperparameter λ is to balance the importance of Neural Auto-correction module in the phase of discrete correction. To dispel your concerns, we conduct sensitivity experiments on two datasets. The results indicate that this module can indeed improve the performance of our method, and the experimental results are robust to hyperparameter λ.
>
> | | λ=0|λ=0.1|λ=0.2|λ=0.5|λ=1.0|
> | - | - | - | - | - | - |
> |  Navier-Stokes(In-t)|3.244E-03|3.017E-03|2.925E-03|2.832E-03| 2.964E-03|
> |  Navier-Stokes(Ext-t)|6.635E-03|6.172E-03|5.873E-03 | 5.764E-03| 5.828E-03|
> |  Prometheus(In-t)|3.623E-03|3.542E-03 | 3.495E-03 | 3.374E-03| 3.545E-03 |
> |  Prometheus(Ext-t)|7.016E-03|6.823E-03|6.747E-03|6.678E-03| 6.837E-03 |

---

> > ### Comment · Reviewer_vtdp · 2025-08-08
> >
> > Thanks for your detailed response. Most of my concerns are well clarified and addressed. I tend to appreciate the authors' argument of the novelty. Thus, I will raise my rating to borderline accept.

---

> ### Comment · Area_Chair_i2xR · 2025-08-05
>
> Dear Reviewer,
>
> As the author-reviewer discussion period is drawing to a close, we kindly ask that you respond to the authors' rebuttal.
>
> Thank you for your work.
>
> Best regards,
> Your AC

---

### Note · Authors · 2025-08-12

Dear reviewers, ACs, and SACs:

We are grateful for the constructive discussion phase! We are pleased to note that all reviewers have confirmed that their concerns have been resolved, and all have clarified they will award positive scores in support of our paper.

In our detailed rebuttals, we have addressed the following key concerns:

- In response to feedback on **Motivation and Novelty**, we elaborated on the distinct roles of each proposed module and the synergistic effects of their integration, which was consistently acknowledged by the reviewers.

- Regarding **Methodological Details**, we provided comprehensive specifics on: (1) the model's training framework and strategy, (2) the purpose of the neural auto-correction mechanism, and (3) the role of key hyperparameters, et al. To further validate our approach, we have added new experiments on hyperparameter sensitivity and 3D scene performance.

- In response to **Model Efficiency and Inference Cost**, we conducted fair efficiency comparison experiments.  The results confirm that our model achieves better performance while maintaining efficiency comparable to the baseline models.

- Regarding the feedback on **Dataset Description**, we provided a more detailed description for both the simulated and real-world data scenarios.

We will definitely incorporate these into our revised version. Thank you again for your hard work during the discussion stage!

Best regards,

the Authors

---

### Decision · Program_Chairs · 2025-09-17

**Decision:**

Accept (poster)

**Comment:**

The authors convincingly address concerns about scalability and efficiency: ODE integration is amortized over the latent grid (not per query), efficiency is competitive with prior work, and new 3D experiments demonstrate favorable memory/runtime scaling. Methodological clarity improved substantially through the discussion: the roles of ODE vs. NAC, the training objective (learning a universal evolution operator from many trajectories), hyperparameter handling (learned μ/γ, robust λ), and the mapping from irregular sensors to a fixed latent grid are now well articulated. The empirical study is broad and adds SSIM/PSNR alongside MSE, with CoPS consistently outperforming baselines in both pointwise and structural fidelity, particularly under sparse observation and long-horizon settings; data partitioning for both simulated (fixed Reynolds, varied initial conditions) and real datasets (ERA5 splits) is now clearly specified.

The primary weakness is limited component-level novelty: several modules build on known ideas, but the integration is coherent and greater than the sum of its parts, and the added analyses/ablation/sensitivity studies are insightful. Given the strong evidence of effectiveness, clarified methodology, scalability results in 3D, and positive reviewer updates (one accept, two borderline accepts after rebuttals), I judge that reasons to accept outweigh reasons to reject.